# Low-rank Interaction with Sparse Additive Effects Model for Large Data Frames

**Geneviève Robin**
Centre de Mathématiques Appliquées
École Polytechnique, XPOP, INRIA
91120 Palaiseau, France
genevieve.robin@polytechnique.edu

**Hoi-To Wai**
Department of SE&EM
The Chinese University of Hong Kong
Shatin, Hong Kong
htwai@se.cuhk.edu.hk

**Julie Josse**
Centre de Mathématiques Appliquées
École Polytechnique, XPOP, INRIA
91120 Palaiseau, France
julie.josse@polytechnique.edu

**Olga Klopp**
ESSEC Business School
CREST, ENSAE
95021 Cergy, France
klopp@essec.edu

**Éric Moulines**
Centre de Mathématiques Appliquées
École Polytechnique, XPOP, INRIA
91120 Palaiseau, France
eric.moulines@polytechnique.edu

## Abstract

Many applications of machine learning involve the analysis of large data frames – matrices collecting heterogeneous measurements (binary, numerical, counts, etc.) across samples – with missing values. Low-rank models, as studied by Udell et al. [27], are popular in this framework for tasks such as visualization, clustering and missing value imputation. Yet, available methods with statistical guarantees and efficient optimization do not allow explicit modeling of main additive effects such as row and column, or covariate effects. In this paper, we introduce a *low-rank interaction and sparse additive effects* (LORIS) model which combines matrix regression on a dictionary and low-rank design, to estimate main effects and interactions simultaneously. We provide statistical guarantees in the form of upper bounds on the estimation error of both components. Then, we introduce a *mixed coordinate gradient descent* (MCGD) method which provably converges sub-linearly to an optimal solution and is computationally efficient for large scale data sets. We show on simulated and survey data that the method has a clear advantage over current practices.

## 1   Introduction

Recently, a lot of effort has been devoted towards the efficient analysis of large data frames, a term coined by Udell et al. [27]. A data frame is a large table of heterogeneous data (binary, numerical, counts) with missing entries, where each row represents an example and each column a feature. In order to analyze them, a powerful technique is to use *low-rank models* that embed rows and columns of data frames into low-dimensional spaces [15, 25, 27], enabling effective data analytics such as clustering, visualization and missing value imputation; see also [18] and the references therein.

Characterizing additive effects of side information – such as covariates, row or column effects – *simultaneously* with low rank interactions is an important extension to plain low-rank models. For example, in data frames obtained from recommender systems, user information and item characteristics are known to influence the ratings in addition to interactions between users and items [7]. These modifications to the low rank model have been advocated in the statistics literature, but they have been implemented only for small data frames [1].

In the large-scale low-rank matrix estimation literature, available methods either do not take additive effects into account [6, 20, 27, 22, 8], or only handle the numerical data [12, 11]. As a common heuristics for preprocessing, prior work such as [20, 27] remove the row and column means and apply some normalization of the row and column variance. We show in numerical experiments this apparently benign operation is not appropriate for large and heterogenous data frames, and can cause severe impairments in the analysis.

The present work investigates a generalization of previous contributions in the analysis of data frames. Our contributions can be summarized as follows.

**Contributions**   We present a new framework that is *statistically* and *computationally* efficient for analyzing large and incomplete heterogeneous data frames.

- We describe in Section 2 the *low-rank interaction with sparse additive effects* (LORIS) model, which combines matrix regression on a dictionary with low rank approximation. We propose a convex doubly penalized quasi-maximum likelihood approach, where the rank constraint is relaxed with a nuclear norm penalty, to estimate the regression coefficients and the low rank component simultaneously. We establish non-asymptotic upper bounds on the estimation errors.

- We propose in Section 3 a Mixed Coordinate Gradient Descent (MCGD) method to solve efficiently the LORIS estimation problem. It uses a mixed update strategy including a proximal update for the sparse component and a conditional gradient (CG) for the low-rank component. We show that the MCGD method converges to an $\epsilon$-optimal solution in $\mathcal{O}(1/\epsilon)$ iterations. We also outline an extension to efficient distributed implementation.

- We demonstrate in Section 4 the efficacy of our method both in terms of estimation and imputation quality on simulated and survey data examples.

**Related work**   Our statistical model and analysis are related to prior work on *low-rank plus sparse matrix decomposition* [28, 3, 4, 13, 17]; these papers provide statistical results for a particular case where the loss function is quadratic and the sparse component is entry-wise sparse. In comparison, the originality of the present work is two-fold. First, the sparsity pattern of the main effects is not restricted to entry-wise sparsity. Second, the data fitting term is not quadratic, but a heterogeneous exponential family quasi log-likelihood. This new framework enables us to tackle many more data sets combining heterogeneous data, main effects and interactions.

For the algorithmic development, our proposed method is related to the prior work such as [21, 26, 5, 11, 29, 14, 24, 9, 19, 2, 10]. These are based on various first-order optimization methods and shall be reviewed in detail in Section 3. Among others, the MCGD method is mostly related to the recent FW-T method by Mu et al. [24] that uses a mixed update rule to tackle a similar estimation problem. There are two differences: first, FW-T is focused on a quadratic loss which is a special case of the statistical estimation problem that we analyze; second, the per-iteration complexity of MCGD is lower as the update rules are simpler. Despite the simplifications, using a new proof technique, we prove that the convergence rate of MCGD is strictly faster than FW-T.

**Notations:**   For any $m \in \mathbb{N}$, $[m] := \{1,...,m\}$. The operator $\mathcal{P}_\Omega(\cdot) : \mathbb{R}^{n \times p} \to \mathbb{R}^{n \times p}$ is the projection operator on the set of entries in $\Omega \subset [n] \times [p]$, and $(\cdot)_+ : \mathbb{R} \to \mathbb{R}_+$ is the projection operator on the non-negative orthant $(x)_+ := \max\{0, x\}$. For matrices, we denote by $\|\cdot\|_F$ the Frobenius norm, $\|\cdot\|_\star$ the nuclear norm, $\|\cdot\|$ the operator norm, and $\|\cdot\|_\infty$ the entry-wise infinity norm. For vectors, we denote by $\|\cdot\|_1$ is the $\ell_1$-norm, $\|\cdot\|_2$ the Euclidean norm, $\|\cdot\|_\infty$ the infinity norm, and $\|\cdot\|_0$ the number of non zero coefficients. The binary operator $\langle \boldsymbol{X}, \boldsymbol{Y} \rangle$ denotes the Frobenius inner product. A function $f : \mathbb{R}^q \to \mathbb{R}$ is said to be $\sigma$-smooth if $f$ is continuously differentiable and $\|\nabla f(\boldsymbol{\theta}) - \nabla f(\boldsymbol{\theta}')\|_2 \leq \sigma \|\boldsymbol{\theta} - \boldsymbol{\theta}'\|_2$ for all $\boldsymbol{\theta}, \boldsymbol{\theta}' \in \mathbb{R}^q$.

## 2   Problem Formulation

**Heterogenous Data Model**   Let $(\mathsf{Y}, \mathsf{X})$ be a probability space equipped with a $\sigma$-finite measure $\mu$. The canonical exponential family distribution $\{\mathrm{Exp}_{h,g}(m), m \in \mathsf{X}\}$ with base measure $h : \mathsf{Y} \to \mathbb{R}^+$, link function $g : \mathsf{X} \to \mathbb{R}$, and scalar parameter, $m \in \mathsf{X}$, has a density given by

$$f_m(y) = h(y) \exp\left(ym - g(m)\right) . \tag{1}$$

The exponential family is a flexible framework to model different types of data. For example, $(\mathsf{Y} = \mathbb{R}, g(m) = m^2\sigma^2/2, h(y) = (2\pi\sigma^2)^{-1/2}\exp(-y^2/2\sigma^2))$ yields a Gaussian distribution with mean $m$ and variance $\sigma^2$ for numerical data; $(\mathsf{Y} = \{0,1\}, g(m) = \log(1 + \exp(m)), h(y) = 1)$ yields a Bernoulli distribution with success probability $1/(1 + \exp(-m))$ for binary data; $(\mathsf{Y} = \mathbb{N}, g(m) = \exp(am), h(y) = 1/y!)$ where $a \in \mathbb{R}$ yields a Poisson distribution with intensity $\exp(am)$ for count data. In these cases, the parameter space is $\mathsf{X} = \mathbb{R}$.

Let $\{(\mathsf{Y}_j, g_j, h_j), \; j \in [p]\}$ be a collection of observation spaces, base and link functions corresponding to the column types of a data frame $\mathbf{Y} = [\mathbf{Y}_{ij}]_{(i,j) \in [n] \times [p]} \in \mathsf{Y}_1^n \times \ldots \times \mathsf{Y}_p^n$. For each $i \in [n]$ and $j \in [p]$, we denote by $\mathbf{M}_{ij}^0$ the target parameter minimizing the Kullback-Leibler divergence between the distribution of $\mathbf{Y}_{ij}$ and the exponential family $\mathrm{Exp}_{h_j, g_j}, j \in [p]$, given by

$$\mathbf{M}_{ij}^0 = \underset{m}{\arg\max} \; \mathbb{E}_{\mathbf{Y}_{ij}}[\log(h_j(\mathbf{Y}_{ij})) + \mathbf{Y}_{ij}m - g_j(m)] . \tag{2}$$

We propose the following model to estimate $\mathbf{M}^0 = [\mathbf{M}_{ij}^0]_{(i,j) \in [n] \times [p]}$ in the presence of additive effects and interactions.

**LOw-rank Interaction with Sparse additive effects (LORIS) model**   For every entry $\mathbf{Y}_{ij}$, assume a vector of covariates $\mathbf{x}_{ij} \in \mathbb{R}^q$ is also available, e.g., user information and item characteristics. Denote $\mathbf{x}_{ij}(k), k \in [q]$ the $k$-th component of $\mathbf{x}_{ij}$ and define the matrix $\mathbf{X}(k) = [\mathbf{x}_{ij}(k)]_{(i,j) \in [n] \times [p]}$. We introduce the following decomposition of the parameter matrix $\mathbf{M}^0$:

$$\mathbf{M}^0 = \sum_{k=1}^{q} \boldsymbol{\alpha}_k^0 \mathbf{X}(k) + \boldsymbol{\Theta}^0. \tag{3}$$

We call (3) the LORIS model, where $\boldsymbol{\alpha} \in \mathbb{R}^q$ is a sparse vector with unknown support modeling additive effects and $\boldsymbol{\Theta}^0 \in \mathbb{R}^{n \times p}$ a low-rank matrix modeling the interactions.

In fact, LORIS is a generalization of *robust* matrix completion [3], where the parameter matrix can be decomposed as the sum of two matrices, one is low-rank and the other has some complementary low-dimensional structure such as entry-wise or column-wise sparsity. Statistical recoverability results in robust matrix estimation under a noiseless setting can be found in [28, 3, 4, 13]; the additive noise setting can be found in a recent work [17]. [23] also provide exact recovery results for more general sparsity patterns.

**Estimation Problem**   Denote $\Omega = \{(i,j) \in [n] \times [p] : \mathbf{Y}_{ij} \text{ is observed}\}$ as the observation set. For $\mathbf{M} \in \mathbb{R}^{n \times p}$, $\mathcal{L}(\mathbf{M})$ is the negative log-likelihood of the observed data $(\mathbf{Y}, \Omega)$ parameterized by $\mathbf{M}$. Up to an additive constant,

$$\mathcal{L}(\mathbf{M}) = \sum_{(i,j) \in \Omega} \{-\mathbf{Y}_{ij}\mathbf{M}_{ij} + g_j(\mathbf{M}_{ij})\} . \tag{4}$$

For $a > 0$, we consider the following estimation problem:

$$(\hat{\boldsymbol{\alpha}}, \hat{\boldsymbol{\Theta}}) \in \underset{\substack{\|\boldsymbol{\alpha}\|_\infty \le a \\ \|\boldsymbol{\Theta}\|_\infty \le a}}{\arg\min} \; \mathcal{L}\left(\sum_{k=1}^{q} \boldsymbol{\alpha}_k \mathbf{X}(k) + \boldsymbol{\Theta}\right) + \lambda_S \|\boldsymbol{\alpha}\|_1 + \lambda_L \|\boldsymbol{\Theta}\|_\star . \tag{5}$$

We denote by $\hat{\mathbf{M}} = \sum_{k=1}^{q} \hat{\boldsymbol{\alpha}}_k \mathbf{X}(k) + \hat{\boldsymbol{\Theta}}$ the estimated parameter matrix. The $\ell_1$ and nuclear norm penalties are convex relaxations of the sparsity and low-rank constraints, and the regularization parameters $\lambda_S$ and $\lambda_L$ serve as trade-offs between fitting the data and enforcing sparsity of $\boldsymbol{\alpha}$ and controlling the "effective rank" of $\boldsymbol{\Theta}$.

**Statistical Guarantees** Here we establish convergence rates for the joint estimation of $\boldsymbol{\alpha}^0$ and $\boldsymbol{\Theta}^0$; the proofs can be found in the supplementary material. Consider the following assumptions.

**H1** $\left\|\boldsymbol{\Theta}^0\right\|_\infty \leq a$, $\left\|\boldsymbol{\alpha}^0\right\|_\infty \leq a$ and for all $k \in [q]$ such that $\alpha_k^0 \neq 0$, $\langle \boldsymbol{\Theta}^0, \mathbf{X}(k) \rangle = 0$.

In particular, H1 guarantees the uniqueness of the decomposition in the LORIS model (3).

**H2** *For $\nu > 0$, all $k \in [q]$ and $(i, j) \in [n] \times [p]$, $\mathbf{X}(k)_{ij} \in [-1, 1]$. Furthermore for all $(i, j) \in [n] \times [p]$, $\sum_{k=1}^q |\mathbf{X}(k)_{ij}| \leq \nu$.*

In particular, H2 guarantees that for all $(\boldsymbol{\Theta}, \boldsymbol{\alpha})$ satisfying H1, the matrix $\mathbf{M} = \sum_{k=1}^q \alpha_k \mathbf{X}(k) + \boldsymbol{\Theta}$ satisfies $\|\mathbf{M}\|_\infty \leq (1 + \nu)a$. Let $\mathbf{G}$ be the $q \times q$ Gram matrix of the dictionary $(\mathbf{X}(1), \ldots, \mathbf{X}(q))$ defined by $\mathbf{G} = [\langle \mathbf{X}(k), \mathbf{X}(l) \rangle]_{(k,l) \in [q] \times [q]}$.

**H3** *For $\kappa > 0$ and all $\boldsymbol{\alpha} \in \mathbb{R}^q$, $\boldsymbol{\alpha}^\top \mathbf{G} \boldsymbol{\alpha} \geq \kappa^2 \|\boldsymbol{\alpha}\|_2^2$.*

Note we do not consider the case where the Gram matrix is singular, e.g., $q > np$. For $0 < \sigma_- \leq \sigma_+ < +\infty$ and $0 < \gamma < \infty$ consider the following assumption on the link functions $g_j$:

**H4** *The functions $g_j$ are twice differentiable, and for all $x \in [-(1 + \nu)a - \gamma, (1 + \nu)a + \gamma]$,*

$$\sigma_-^2 \leq g_j''(x) \leq \sigma_+^2, \ j \in [p].$$

H4 implies the data fitting term $\mathcal{L}(\mathbf{M})$ is smooth and satisfies a restricted strong convexity property.

**H5** *For all $(i, j) \in [n] \times [p]$, $Y_{ij}$ is a sub-exponential random variable with scale and variance parameters $1/\gamma$ and $\sigma_+^2$.*

If the random variables $Y_{ij}$ are actually distributed according to an exponential family distribution of the form (1), then H4 implies H5.

**H6** *For $(i, j) \in [n] \times [p]$, the events $\omega_{ij} = \{(i, j) \in \Omega\}$ are independent with occurrence probability $\pi_{ij}$. Furthermore, there exists $0 < \pi \leq 1$ such that for all $(i, j) \in [n] \times [p]$, $\pi_{ij} \geq \pi$.*

H6 implies a data missing-at-random scenario where $\mathbf{Y}_{ij}$ is observed with probability at least $\pi$.

**Theorem 1** *Assume H1-6. Set*

$$\lambda_L = 2C\sigma_+ \sqrt{\pi \max(n, p) \log(n + p)}, \ \text{and} \ \lambda_S = 24 \max_k \|\mathbf{X}(k)\|_1 \log(n + p)/\gamma, \quad (6)$$

*where $C$ is a positive constant. Assume that $\max(n, p) \geq 4\sigma_+^2/\gamma^6 \log^2(\sqrt{\min(n, p)/(\pi \gamma \sigma_-)}) + 2 \exp(\sigma_+^2/\gamma^2 + 2\sigma_+^2 \gamma a)$. Then, with probability at least $1 - 9(n + p)^{-1}$,*

$$
\begin{aligned}
\left\|\hat{\boldsymbol{\alpha}} - \boldsymbol{\alpha}^0\right\|_2^2 &\leq C_1 \frac{s \max_k \|\mathbf{X}(k)\|_1 \log(n + p)}{\kappa^2 \pi} + \mathsf{D}_{\boldsymbol{\alpha}}, \\
\left\|\hat{\boldsymbol{\Theta}} - \boldsymbol{\Theta}^0\right\|_F^2 &\leq C_2 \left( \frac{r \max(n, p)}{\pi} + \frac{s \max_k \|\mathbf{X}(k)\|_1}{\pi} \right) \log(n + p) + \mathsf{D}_{\boldsymbol{\Theta}}.
\end{aligned}
\quad (7)
$$

*In (7), $s := \|\boldsymbol{\alpha}^0\|_0$, $r := \mathrm{rank}(\boldsymbol{\Theta}^0)$. $C_1$ and $C_2$ are positive constants and $\mathsf{D}_{\boldsymbol{\alpha}}$ and $\mathsf{D}_{\boldsymbol{\Theta}}$ are residuals of lower order whose exact values are given in Appendix A.*

The proof can be found in Appendix A. In Theorem 1, the rate obtained for $\boldsymbol{\alpha}^0$ is the same as the bound obtained in [17] in the special case of robust matrix completion. Examples satisfying $\max_k \|\mathbf{X}(k)\|_1 / \kappa^2 = \mathcal{O}(1)$ include the case where the elements of the dictionary are matrices are all zeros except a row or a column of one, (to model row and column effects) and the number of rows $n$ and columns $p$ are of the same order; or when the covariates $\mathbf{x}_{ij}$ are categorical and the categories are balanced, *i.e.,* the number of samples per category is of the same order.

The rate obtained for $\boldsymbol{\Theta}^0$ is the sum of the standard low-rank matrix completion rate of order $r \max(n, p)/\pi$, e.g., [16], and of a term which boils down to sparse vector estimation rate as long as $\max_k \|\mathbf{X}(k)\|_1 = \mathcal{O}(1)$. Again, the latter can be satisfied by the special case of robust matrix completion, for which our rates match the results of [17].

# 3 A Mixed Coordinate Gradient Descent Method for LORIS

This section introduces a mixed coordinate gradient descent (MCGD) method to solve the LORIS estimation problem (5). We assume that $a$ is sufficiently large such that the constraints $\|\boldsymbol{\alpha}\|_\infty \leq a, \|\boldsymbol{\Theta}\|_\infty \leq a$ are always inactive. To simplify notation, we denote the log-likelihood function as $\mathcal{L}(\boldsymbol{\alpha}, \boldsymbol{\Theta}) := \mathcal{L}\left(\sum_{k=1}^q \boldsymbol{\alpha}_k \mathbf{X}(k) + \boldsymbol{\Theta}\right)$. We assume

**H7** *(a) $\mathcal{L}(\boldsymbol{\alpha}, \boldsymbol{\Theta})$ is $\sigma_{\boldsymbol{\Theta}}$-smooth w.r.t. $\Theta_{ij}$ for $(i,j) \in \Omega$ and (b) $\sigma_{\boldsymbol{\alpha}}$-smooth w.r.t. $\boldsymbol{\alpha}$; (c) the gradient $\nabla_{\boldsymbol{\alpha}} \mathcal{L}(\boldsymbol{\alpha}, \boldsymbol{\Theta})$ is $\hat{\sigma}_{\boldsymbol{\Theta}}$-Lipschitz w.r.t. $\boldsymbol{\Theta}$. Moreover, the gradient $\nabla_{\boldsymbol{\Theta}} \mathcal{L}(\boldsymbol{\alpha}, \boldsymbol{\Theta})$ is bounded as long as $\boldsymbol{\alpha}, \boldsymbol{\Theta}$ are bounded.*

The above is implied by H4 for bounded $(\boldsymbol{\alpha}, \boldsymbol{\Theta})$. We consider the augmented objective function:

$$F(\boldsymbol{\alpha}, \boldsymbol{\Theta}, R) := \mathcal{L}(\boldsymbol{\alpha}, \boldsymbol{\Theta}) + \lambda_S \|\boldsymbol{\alpha}\|_1 + \lambda_L R . \tag{8}$$

For some $R_{\mathsf{UB}} \geq 0$, if an optimal solution $(\hat{\boldsymbol{\alpha}}, \hat{\boldsymbol{\Theta}})$ to (5) satisfies $\|\hat{\boldsymbol{\Theta}}\|_\star \leq R_{\mathsf{UB}}$, then any optimal solution to the following problem

$$\mathsf{P}(R_{\mathsf{UB}}) : \quad \min_{\boldsymbol{\alpha} \in \mathbb{R}^q, \boldsymbol{\Theta} \in \mathbb{R}^{n \times p}, R \in \mathbb{R}_+} F(\boldsymbol{\alpha}, \boldsymbol{\Theta}, R) \text{ s.t. } R_{\mathsf{UB}} \geq R \geq \|\boldsymbol{\Theta}\|_\star , \tag{9}$$

will also be optimal to (5). For example, $(\hat{\boldsymbol{\alpha}}, \hat{\boldsymbol{\Theta}}, \hat{R})$ with $\hat{R} = \|\hat{\boldsymbol{\Theta}}\|_\star$ is an optimal solution to (9). We have defined the problem as $\mathsf{P}(R_{\mathsf{UB}})$ to emphasize its dependence on the upper bound $R_{\mathsf{UB}}$. Later we shall describe a simple strategy to estimate $R_{\mathsf{UB}}$. We fix the set $\Xi \subseteq [n] \times [p]$ where $\Omega \subseteq \Xi$ is the target coordinate set for the low rank matrix $\hat{\boldsymbol{\Theta}}$ that we are interested in.

**Proposed Method** A natural way to exploit structure in $\mathsf{P}(R_{\mathsf{UB}})$ is to apply coordinate gradient descent to update $\boldsymbol{\alpha}$ and $(\boldsymbol{\Theta}, R)$ separately. While the trace-norm constraint on $(\boldsymbol{\Theta}, R)$ can be handled by the conditional gradient (CG) method [14], the $\ell_1$ norm penalization on $\boldsymbol{\alpha}$ is more efficiently tackled by the proximal gradient method in practice. In addition, we tighten the upper bound $R_{\mathsf{UB}}$ on-the-fly as the algorithm proceeds. The MCGD method goes as follows. At the $t$th iteration, we are given the previous iterate $(\boldsymbol{\alpha}^{(t-1)}, \boldsymbol{\Theta}^{(t-1)}, R^{(t-1)})$ and the upper bound $R_{\mathsf{UB}}^{(t)}$ is computed. The first block $\boldsymbol{\alpha}$ is updated with a proximal gradient step:

$$
\begin{aligned}
\boldsymbol{\alpha}^{(t)} &= \mathrm{prox}_{\gamma\lambda_S \|\cdot\|_1}\left(\boldsymbol{\alpha}^{(t-1)} - \gamma \nabla_{\boldsymbol{\alpha}} \mathcal{L}(\boldsymbol{\alpha}^{(t-1)}, \boldsymbol{\Theta}^{(t-1)})\right) \\
&= \mathsf{T}_{\gamma\lambda_S}\left(\boldsymbol{\alpha}^{(t-1)} - \gamma \nabla_{\boldsymbol{\alpha}} \mathcal{L}(\boldsymbol{\alpha}^{(t-1)}, \boldsymbol{\Theta}^{(t-1)})\right) .
\end{aligned}
\tag{10}
$$

In (10), $\nabla_{\boldsymbol{\alpha}} \mathcal{L}(\cdot)$ is the gradient of the log-likelihood function taken *w.r.t.* $\boldsymbol{\alpha}$, $\gamma > 0$ is a pre-defined step size parameter and $\mathsf{T}_\lambda(\boldsymbol{x}) := \mathrm{sign}(\boldsymbol{x}) \odot (\boldsymbol{x} - \lambda\mathbf{1})_+$ is the component-wise soft thresholding operator. Alternatively, we can exactly solve the problem

$$\boldsymbol{\alpha}^{(t)} \in \arg\min_{\boldsymbol{\alpha} \in \mathbb{R}^q} F(\boldsymbol{\alpha}, \boldsymbol{\Theta}^{(t-1)}, R^{(t-1)}) , \tag{11}$$

for which closed-form solution can be obtained in certain special cases (see below).

The second block $(\boldsymbol{\Theta}, R)$ is updated with a CG step

$$(\boldsymbol{\Theta}^{(t)}, R^{(t)}) = (\boldsymbol{\Theta}^{(t-1)}, R^{(t-1)}) + \beta_t (\hat{\boldsymbol{\Theta}}^{(t)} - \boldsymbol{\Theta}^{(t-1)}, \hat{R}^{(t)} - R^{(t-1)}) , \tag{12}$$

where $\beta_t \in [0, 1]$ is a step size to be defined later. $(\hat{\boldsymbol{\Theta}}^{(t)}, \hat{R}^{(t)})$ is a direction evaluated as

$$(\hat{\boldsymbol{\Theta}}^{(t)}, \hat{R}^{(t)}) \in \arg\min_{\boldsymbol{Z}, R} \langle \boldsymbol{Z}, \nabla_{\boldsymbol{\Theta}} \mathcal{L}(\boldsymbol{\alpha}^{(t)}, \boldsymbol{\Theta}^{(t-1)})\rangle + \lambda_1 R \text{ s.t. } \|\boldsymbol{Z}\|_\star \leq R \leq R_{\mathsf{UB}}^{(t)} , \tag{13}$$

and $\nabla_{\boldsymbol{\Theta}} \mathcal{L}(\cdot)$ is the gradient of $\mathcal{L}(\cdot)$ taken *w.r.t.* $\boldsymbol{\Theta}$. If $(\boldsymbol{\Theta}^{(t-1)}, R^{(t-1)})$ is feasible to $\mathsf{P}(R_{\mathsf{UB}}^{(t)})$, then $(\boldsymbol{\Theta}^{(t)}, R^{(t)})$ must also be feasible to $\mathsf{P}(R_{\mathsf{UB}}^{(t)})$. Furthermore, if we let $\boldsymbol{u}_1, \boldsymbol{v}_1$ be the top left and right singular vectors of the gradient matrix $\nabla_{\boldsymbol{\Theta}} \mathcal{L}(\boldsymbol{\alpha}^{(t)}, \boldsymbol{\Theta}^{(t-1)})$ and $\sigma_1(\nabla_{\boldsymbol{\Theta}} \mathcal{L}(\boldsymbol{\alpha}^{(t)}, \boldsymbol{\Theta}^{(t-1)}))$ be the top singular value, then $(\hat{\boldsymbol{\Theta}}^{(t)}, \hat{R}^{(t)})$ admits a simple closed form solution:

$$(\hat{\boldsymbol{\Theta}}^{(t)}, \hat{R}^{(t)}) = \begin{cases} (\mathbf{0}, 0), & \text{if } \lambda_L \geq \sigma_1(\nabla_{\boldsymbol{\Theta}} \mathcal{L}(\boldsymbol{\alpha}^{(t)}, \boldsymbol{\Theta}^{(t-1)})) , \\ (-R_{\mathsf{UB}}^{(t)} \boldsymbol{u}_1 \boldsymbol{v}_1^\top, R_{\mathsf{UB}}^{(t)}), & \text{if } \lambda_L < \sigma_1(\nabla_{\boldsymbol{\Theta}} \mathcal{L}(\boldsymbol{\alpha}^{(t)}, \boldsymbol{\Theta}^{(t-1)})) . \end{cases} \tag{14}$$

Lastly, the step size $\beta_t$ is determined by:

$$\beta_t = \min\left\{1, \frac{\langle \boldsymbol{\Theta}^{(t-1)} - \hat{\boldsymbol{\Theta}}^{(t)}, \nabla_{\boldsymbol{\Theta}}\mathcal{L}(\boldsymbol{\alpha}^{(t)}, \boldsymbol{\Theta}^{(t-1)})\rangle + \lambda_L(R^{(t-1)} - \hat{R}^{(t)})}{\sigma_{\boldsymbol{\Theta}}\|\mathcal{P}_\Omega(\hat{\boldsymbol{\Theta}}^{(t)} - \boldsymbol{\Theta}^{(t-1)})\|_{\mathrm{F}}^2}\right\}. \quad (15)$$

The step size strategy ensures decrease in the objective value between successive iterations. This is essential for establishing convergence of the proposed method [cf. Theorem 2]. We remark that the arithmetics in the MCGD method are not affected when we restrict the update of $\boldsymbol{\Theta}^{(t)}$ in (12) to the entries in $\Xi$ only. This is due to $\mathcal{L}(\boldsymbol{X}) = \mathcal{L}(\mathcal{P}_\Omega(\boldsymbol{X}))$ and the CG update direction (13) only involves the gradient of $\nabla_{\boldsymbol{\Theta}}\mathcal{L}(\boldsymbol{\alpha}^{(t)}, \boldsymbol{\Theta}^{(t-1)})$ *w.r.t.* entries of $\boldsymbol{\Theta}$ in $\Omega$, where $\Omega \subseteq \Xi$.

**Computing the Upper Bound $R_{\mathsf{UB}}^{(t)}$** We describe a strategy for computing a valid upper bound $R_{\mathsf{UB}}^{(t)}$ for $\hat{R}$ and $\|\hat{\boldsymbol{\Theta}}\|_\star$ during the updates in the MCGD method. Let us assume that:

**H8** *For all $\boldsymbol{\Theta}$ and $\boldsymbol{\alpha}$, we have $\mathcal{L}(\boldsymbol{\alpha}, \boldsymbol{\Theta}) \geq 0$.*

The above can be enforced as the log-likelihood function is lower bounded [cf. H4]. From (5) and using the above assumption, it is obvious that

$$F_0(\mathbf{0}, \mathbf{0}) = \mathcal{L}(\mathbf{0}, \mathbf{0}) \geq \mathcal{L}(\hat{\boldsymbol{\alpha}}, \hat{\boldsymbol{\Theta}}) + \lambda_S\|\hat{\boldsymbol{\alpha}}\|_1 + \lambda_L\|\hat{\boldsymbol{\Theta}}\|_\star \geq \lambda_L\|\hat{\boldsymbol{\Theta}}\|_\star, \quad (16)$$

and thus $R_{\mathsf{UB}}^0 := \lambda_L^{-1}\mathcal{L}(\mathbf{0} + f_{\boldsymbol{U}}(\mathbf{0}))$ is a valid upper bound to $\|\hat{\boldsymbol{\Theta}}\|_\star$; furthermore it can be tightened as we progress in the MCGD method. In particular, observe that $(\hat{\boldsymbol{\alpha}}, \hat{\boldsymbol{\Theta}}, \hat{R})$ with $\hat{R} = \|\hat{\boldsymbol{\Theta}}\|_\star$ is an optimal solution to $\mathsf{P}(R_{\mathsf{UB}}^0)$, we have

$$F(\boldsymbol{\alpha}, \boldsymbol{\Theta}, R) \geq F(\hat{\boldsymbol{\alpha}}, \hat{\boldsymbol{\Theta}}, \hat{R}) = \mathcal{L}(\hat{\boldsymbol{\alpha}}, \hat{\boldsymbol{\Theta}}) + \lambda_S\|\hat{\boldsymbol{\alpha}}\|_1 + \lambda_L\hat{R} \geq \lambda_L\hat{R}. \quad (17)$$

In other words, for all feasible $(\boldsymbol{\alpha}, \boldsymbol{\Theta}, R)$ to $\mathsf{P}(R_{\mathsf{UB}}^0)$, $\lambda_L^{-1}F(\boldsymbol{\alpha}, \boldsymbol{\Theta}, R)$ is an upper bound to $\hat{R}$ and $\|\hat{\boldsymbol{\Theta}}\|_\star$. The above motivates us to select $R_{\mathsf{UB}}^{(t)} := \lambda_L^{-1}F(\boldsymbol{\alpha}^{(t)}, \boldsymbol{\Theta}^{(t-1)}, R^{(t-1)})$ at iteration $t$, where we observe that $R_{\mathsf{UB}}^{(t)} \geq R^{(t-1)}$. That is, $(\boldsymbol{\alpha}^{(t)}, \boldsymbol{\Theta}^{(t-1)}, R^{(t-1)})$ is feasible to both $\mathsf{P}(R_{\mathsf{UB}}^{(t)})$ and $\mathsf{P}(R_{\mathsf{UB}}^{(t-1)})$. Lastly, we summarize the MCGD method in Algorithm 1.

**Computation Complexity** Consider the MCGD method in Algorithm 1. Observe that line 3 requires computing the gradient *w.r.t.* $\boldsymbol{\alpha}$ which involves $|\Omega|q$ Floating Points Operations (FLOPS) and the soft thresholding operator involves $\mathcal{O}(q)$ FLOPS. As the log-likelihood function $\mathcal{L}(\cdot)$ is evaluated element-wisely on $\boldsymbol{\Theta}$, evaluating the objective value and the derivative *w.r.t.* $\boldsymbol{\Theta}$ requires $\mathcal{O}(|\Omega|)$ FLOPS. As such, line 4 can be evaluated in $\mathcal{O}(|\Omega|)$ FLOPS and line 5 requires $\mathcal{O}(|\Omega|\max\{n,p\}\log(1/\delta))$ FLOPS where the additional complexity is due to the top SVD computation and $\delta$ is a preset accuracy level of SVD computation. Lastly, line 6 requires $\mathcal{O}(|\Xi|)$ FLOPS since we only need to update the entries of $\boldsymbol{\Theta}$ in $\Xi$ [cf. see the remark after (15)]. The overall per-iteration complexity is $\mathcal{O}(|\Xi| + |\Omega|(\max\{n,p\}\log(1/\delta) + q))$.

---

**Algorithm 1** MCGD Method for (9).

1: **Initialize:** — $\boldsymbol{\Theta}^{(0)}, \boldsymbol{\alpha}^{(0)}, R^{(0)}$. E.g., $\boldsymbol{\Theta}^{(0)}, \boldsymbol{\alpha}^{(0)}, R^{(0)} = (\mathbf{0}, \mathbf{0}, 0)$.
2: **for** $t = 1, 2, \ldots, T$ **do**
3:    *// Update for $\boldsymbol{\alpha}$ //*
   Compute the proximal update using (10) [or exact update via (11)] to obtain $\boldsymbol{\alpha}^{(t)}$.
4:    *// Update for $(\boldsymbol{\Theta}, R)$ //*
   Compute the upper bound as $R_{\mathsf{UB}}^{(t)} := \lambda_L^{-1}F(\boldsymbol{\alpha}^{(t)}, \boldsymbol{\Theta}^{(t-1)}, R^{(t-1)})$.
5:    Compute the update direction, $(\hat{\boldsymbol{\Theta}}^{(t)}, \hat{R}^{(t)})$, using Eq. (14).
6:    Compute the CG update using (12), where the step size $\beta_t$ is set as Eq. (15).
7: **end for**
8: **Return:** $\boldsymbol{\Theta}^{(T)}, \boldsymbol{\alpha}^{(T)}, R^{(T)}$.

---

From the above, the per-iteration computation complexity of the MCGD method scales linearly with the problem dimension $\max\{n, p\}$ and $|\Omega|$. This is comparable to [24, 9], where the former focuses only on the least square loss case. The following theorem, whose proof can be found in Appendix C, shows that the MCGD method converges at a sublinear rate.

**Theorem 2** *Assume H7 and H8. Define the quantity*

$$C(t) := \max\left\{\frac{24(Q^{(t)})^2}{\gamma}, \frac{24\hat{\sigma}_{\boldsymbol{\Theta}}^2(Q^{(t)})^2}{\sigma_{\boldsymbol{\Theta}}} + \max\{6R_{\mathsf{UB}}^{(t)}(\lambda_L + M^{(t)}), 24\sigma_{\boldsymbol{\Theta}}(R_{\mathsf{UB}}^{(t)})^2\}\right\}, \quad (18)$$

where $Q^{(t)} := \lambda_S^{-1} F(\boldsymbol{\alpha}^{(t)}, \boldsymbol{\Theta}^{(t)}, R^{(t)})$, $M^{(t)} := \|\nabla_{\boldsymbol{\Theta}} \mathcal{L}(\boldsymbol{\alpha}^{(t)}, \boldsymbol{\Theta}^{(t-1)})\|_2$ and $R_{\mathrm{UB}}^{(t)} := \lambda_L^{-1} F(\boldsymbol{\alpha}^{(t)}, \boldsymbol{\Theta}^{(t-1)}, R^{(t-1)})$. If we choose the step sizes as $\gamma \leq 1/\sigma_{\boldsymbol{\alpha}}$ and $\beta_t$ as in (15), then (i) the above quantity is upper bounded as $C(t) \leq \overline{C}$ for all $t \geq 1$, where

$$\overline{C} := \max\left\{ \frac{24(Q^{(0)})^2}{\gamma}, \frac{24\hat{\sigma}_{\boldsymbol{\Theta}}^2 (Q^{(0)})^2}{\sigma_{\boldsymbol{\Theta}}} + \max\{6R_{\mathrm{UB}}^{(0)}(\lambda_L + \bar{M}), 24\sigma_{\boldsymbol{\Theta}}(R_{\mathrm{UB}}^{(0)})^2\} \right\}, \qquad (19)$$

such that $\bar{M}$ is an upper bound to $M^{(t)}$, and (ii) the MCGD method converges to an $\epsilon$-optimal solution to (5) in $T$ iterations, i.e., $F_0(\boldsymbol{\alpha}^{(T)}, \boldsymbol{\Theta}^{(T)}) - F_0(\hat{\boldsymbol{\alpha}}, \hat{\boldsymbol{\Theta}}) \leq \epsilon$, where

$$T \geq \overline{C}(T)\left( \frac{1}{\epsilon} - \frac{1}{F_0(\boldsymbol{\alpha}^{(0)}, \boldsymbol{\Theta}^{(0)}) - F_0(\hat{\boldsymbol{\alpha}}, \hat{\boldsymbol{\Theta}})} \right)_+ \quad \text{with} \quad \overline{C}(T) := \left( \frac{1}{T} \sum_{t=1}^{T} \frac{1}{C(t)} \right)^{-1}. \qquad (20)$$

In particular, as $\overline{C}(T) \leq \overline{C}$, at most $\overline{C}(\epsilon^{-1} - (F_0(\boldsymbol{\alpha}^{(0)}, \boldsymbol{\Theta}^{(0)}) - F_0(\hat{\boldsymbol{\alpha}}, \hat{\boldsymbol{\Theta}}))^{-1})_+$ iterations are required for the MCGD method to reach an $\epsilon$-optimal solution to (5).

**Detailed Comparison to Prior Algorithms**    Previous contributions have focused on the special case of (5) where $q = np$, the dictionary $(\mathbf{X}(1), \ldots, \mathbf{X}(q))$ is the canonical basis of $\mathbb{R}^{n \times p}$, and the link functions are quadratic. In this particular case, (5) becomes the estimation problem solved in sparse plus low-rank matrix decomposition. Popular examples are the alternating direction method of multiplier [21, 26] or the projected gradient method on a reformulated problem [5]. These methods either require computing a complete SVD or knowing the optimal rank number of $\boldsymbol{\Theta}$ a priori. When $n, p \gg 1$, it is computationally prohibitive to evaluate the complete SVD since each iteration would require $\mathcal{O}(\max\{n^2 p, p^2 n\})$ FLOPS. Other related work rely on factorizing the low-rank component, yielding nonconvex problems [11]; see also [29] and references therein.

Similar to the development of MCGD, a natural alternative is to apply algorithms based on the CG (a.k.a. Frank-Wolfe) method [14], whose iterations only require the computation of a top SVD. The present work is closely related to the efforts in [24, 9] which focused on the quadratic setting. Mu et al. [24] combines the CG method with proximal update as a two-steps procedure; Garber et al. [9] combines a CD method with CG updates on both the sparse and low-rank components. The work in [9] is also related to [19, 2] which combine CD with CG updates for solving constrained problems, instead of penalized problems like (5). Sublinear convergence rates are proven for the above methods. Finally, Fithian and Mazumder [8] also suggested to apply CD on (5), yet the convergence properties were not discussed.

In fact, when the MCGD's result is specialized to the same setting as [24], our worst-case bound on iteration number computed with $\overline{C}$ match the bound in [24]. As shown in the supplementary material, we have $C(t) \to C^\star$, where $C^\star$ depends on the optimal objective value of (9) and is smaller than $\overline{C}$. Since the quantity $\overline{C}(T)$ in (20) is an average of $\{C(t)\}_{t=1}^{T}$, this implies that the MCGD method requires less number of iterations for convergence than that is required by [24]. Such reduction is possible due to the on-the-fly update for $R_{\mathrm{UB}}^{(t)}$. Moreover, our analysis in Theorem 2 holds when the MCGD method is implemented with a few practical modifications.

**Exact Partial Minimization for $\boldsymbol{\alpha}$**    Consider the special case of (5) where the link functions are either quadratic or exponential and the dictionary matrices satisfy:

$$\mathrm{supp}(\boldsymbol{X}(k)) \cap \mathrm{supp}(\boldsymbol{X}(k')) = \emptyset, \ k \neq k' \ \text{and} \ [\boldsymbol{X}(k)]_{i,j} = c_k, \ \forall \, (i,j) \in \mathrm{supp}(\boldsymbol{X}(k)) . \qquad (21)$$

In this case, the partial minimization (11) can be decoupled into $q$ scalar optimizations involving one coordinate of $\boldsymbol{\alpha}$, which can be solved in closed form. Note that this modification to the MCGD method is supported by Theorem 2 and the sublinear convergence rate holds. On the contrary, closed form update of $\boldsymbol{\alpha}$ is not supported by prior works such as [24, 9, 19, 2].

**Distributed MCGD Optimization**    Consider the case where the observed data entries are stored across $K$ workers, each of them communicating with a central server. It is natural to distribute the MCGD optimization over these workers to offload computation burden, or for privacy protection. Formally, we divide $\Omega$ into $K$ disjoint partitions such that $\Omega = \Omega_1 \cup \cdots \cup \Omega_K$ and worker $k$ holds $\Omega_k$. In this way, $\mathcal{L}(\boldsymbol{\alpha}, \boldsymbol{\Theta}) = \sum_{k=1}^{K} \mathcal{L}_k(\boldsymbol{\alpha}, \boldsymbol{\Theta})$, where $\mathcal{L}_k(\boldsymbol{\alpha}, \boldsymbol{\Theta})$ is defined by replacing the summation over $\Omega$ with $\Omega_k$ in (4). Clearly, when $\boldsymbol{\alpha}$ and $\mathcal{P}_{\Omega_k}(\boldsymbol{\Theta})$ are given to the $k$th worker, the worker will be

able to evaluate the *local* loss function and its gradient.

As shown in Appendix D, the MCGD method can be easily extended to utilize distributed computation. The proximal update in line 3 is replaced by the following procedure. First, the *local* gradients computed by the workers are aggregated, then the soft thresholding operation is performed at the central server. Meanwhile, as the CG update in line 5 essentially requires computing the top singular vectors of the gradient matrix $\nabla_{\Theta} L(\alpha, \Theta) = \sum_{k=1}^{K} \nabla_{\Theta} \mathcal{L}_k(\alpha, \mathcal{P}_{\Omega_k}(\Theta))$, the latter can be implemented through a distributed version of the power method exploiting the decomposable structure of the gradient, such as described in [30]. It only requires $\mathcal{O}(\log(1/\delta))$ power iterations to compute a top SVD solution of accuracy $\delta$. Thus, for a sufficiently small $\delta > 0$, the overall per-iteration complexity of the distributed method at the $t$th iteration is reduced to $\mathcal{O}(|\Xi| + \max\{n, p\} \log(1/\delta))$ at the central server, and $\mathcal{O}(|\Omega_k|(\max\{n, p\} \log(1/\delta) + q))$ at the $k$th worker.

# 4   Numerical Experiments

**Experimental Setup**    We first generate the target parameter $\mathbf{M}^0$ according to the LORIS model in (3). For the sparse additive effects component, we consider $q = pn/5$ where we set $(\mathbf{X}(k))_{ij} = 1$ if $j(n-1) + i \in \{5(k-1) + 1, ..., 5k\}$. This models a categorical variable containing $n/5$ categories. Furthermore, the target sparse component $\alpha^0$ has a sparsity level of $10\%$. For the low-rank component, the target parameter $\Theta^0$ is generated as a rank-4 matrix formed by the outer product of random orthogonal vectors. Notice that due to the structure of sparse additive effects, the surveyed prior methods [21, 11, 5] cannot be applied directly.

**Gaussian Design**    To compare our framework to a reasonable benchmark, we focus on a homogenous setting with numerical data modeled with the quadratic link function $g(m) = m^2$. We set the regularization parameters $\lambda_S$ and $\lambda_L$ to the theoretical values given in Theorem 1. We compare our result with a common two-step procedure where the components $\alpha_{kj}$ are first estimated in a preprocessing step as the means of the variables taken by group; then $\Theta$ is estimated using the softImpute method proposed in [12]. The regularization parameter for [12] is set to the same value $\lambda_L$. We compare the results in terms of estimation error and computing time in Table 1, after letting the two methods converge to the same precision of $10^{-5}$. We observe the two methods perform equally well in terms of estimating $\Theta$. LORIS yields constant estimation errors of $\alpha^0$ as the dimension increases and the support of $\alpha^0$ is kept constant, contrary to the two-step procedure for which the estimation error of $\alpha^0$ increases with the dimension. As expected, the two-step method is faster for small data sets, whereas for large data sizes LORIS is superior in computational time. The above results are consistent with our theoretical findings.

| **problem size** ($n \times p$) | **time** (secs) | | $\left\lVert \Theta^0 - \hat{\Theta} \right\rVert_F^2$ | | $\left\lVert \alpha^0 - \hat{\alpha} \right\rVert_2^2$ | |
|---|---|---|---|---|---|---|
| | LORIS | two-step | LORIS | two-step | LORIS | two-step |
| $150 \times 30$ | 0.17 | 0.02 | 52 | 52 | 1.8 | 3.0 |
| $1,500 \times 300$ | 13.8 | 10.7 | 175.5 | 234 | 0.95 | 17.1 |
| $15,000 \times 300$ | 130.2 | 136.6 | 675 | 720 | 0.95 | 16.2 |
| $15,000 \times 3,000$ | 348 | 528 | $2.7 \times 10^3$ | $2.6 \times 10^3$ | 2.34 | 180 |

Table 1: Comparison of proposed method with a two-step method in terms of computation time and estimation error for increasing dimensions (averaged over 10 experiments).

**Survey data**    To test the efficacy of our framework with heterogeneous data, we examine a survey conducted by the French National Institute of Statistics (Insee: http://www.insee.fr/) concerning the hobbies of French people. The data set contains $n = 8,403$ individuals and $p = 19$ binary and quantitative variables, indicating whether or not the person has been involved in different activities (reading, fishing, etc.), the number of hours spent watching TV and the overall number of hobbies of the individuals. Individuals are grouped by age category ($15 - 25$, $25 - 35$, etc.): this categorical variable is used as a predictor of the survey responses in the subsequent experiment. We introduce $30\%$ of missing values in the data set, and compare the imputation error of LORIS with a mixed data model (using a quadratic loss for numeric columns, a logistic loss for binary columns and a Poisson loss for counts) and LORIS with a Gaussian data model, with the imputation error of softImpute. The

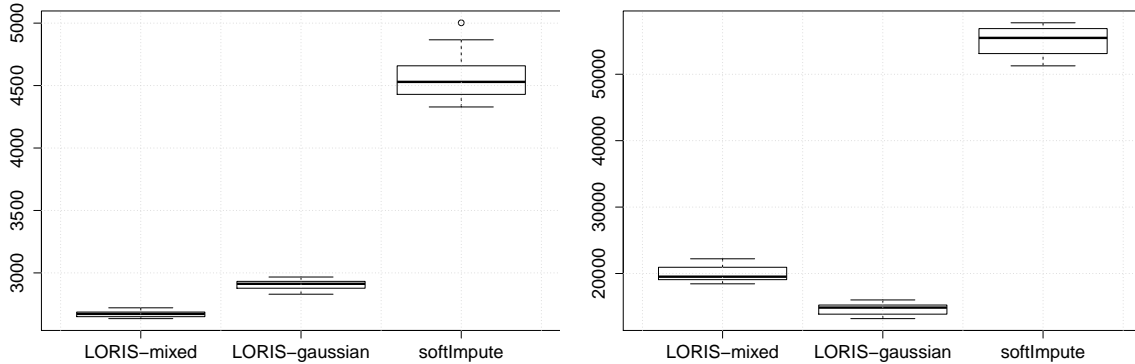

Figure 2: Imputation error of LORIS with mixed data model and Gaussian data model, and softImpute (10 replications) for categorical variables (left) and quantitative variables (right).

results are given in Figure 2 across 10 replications of the experiment, and show that, for this example, both LORIS models improve on the baseline softImpute by a factor 2. We also observe that modeling explicitly the binary variables leads to better imputation.

Finally, we apply LORIS with a mixed data model to the original data set. A subset of the resulting $\boldsymbol{\alpha}$ vector is given in Table 2. There is a coefficient in $\boldsymbol{\alpha}_{kj}$ for every age category $k$ and every variable $j$. The coefficients in Table 2 indicate that young individuals engage in activities such as music and sport more than older people, and the opposite trend for collecting, knitting and fishing. Some coefficients are set to zero, indicating the absence of effect of the age category on the variable. We also observe that younger people engage overall in more activities than older people.

| Age category | Music | Sport | Collecting | Mechanic | Knitting | Fishing | Nb activities |
|---|---|---|---|---|---|---|---|
| 25-35 | 2.2 | 0.4 | -2.1 | 0 | -1.7 | -1.9 | 10.0 |
| 35-45 | 2.0 | 0.3 | -2.7 | 0 | -2.3 | -2.3 | 13.0 |
| 45-55 | 1.1 | -0.8 | -2.1 | 0 | -2.7 | -2.7 | 13.8 |
| 55-65 | 0 | -2.2 | -1.9 | 0 | -1.0 | -1.6 | 8.8 |
| 65-75 | 0 | -2.1 | -1.4 | -1.1 | -0.7 | -1.3 | 5.5 |
| 75-85 | -0.1 | -0.9 | -0.6 | -0.5 | -0.1 | -0.6 | 2.2 |

Table 2: Estimated age category effects ($\boldsymbol{\alpha}$).

**Conclusion** In this paper, we proposed a new framework for handling large data frames with heterogeneous data and missing values which incorporates additive effects. It consists of a doubly penalized quasi-maximum likelihood estimator and a new optimization algorithm to implement the estimator. We examined both the statistical and computational efficiency of the framework and derived worst case bounds of its performance. Future work includes the incorporation of qualitative features with more than two categories and of missing values in the dictionary matrices.

## 5 Acknowledgement

The authors would like to thank for the useful comments from three anonymous reviewers. HTW's work was supported by the grant NSF CCF-BSF 1714672.

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
