[Supplementary Material]

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

 \mathbf{\Theta}^{(t-1)} - \hat{\mathbf{\Theta}}^{(t)}, \nabla_{\mathbf{\Theta}}\mathcal{L}(\boldsymbol{\alpha}^{(t)}, \mathbf{\Theta}^{(t-1)})\rangle + \lambda_L(R^{(t-1)} - \hat{R}^{(t)})}{\sigma_{\mathbf{\Theta}}\|\mathcal{P}_\Omega(\hat{\mathbf{\Theta}}^{(t)} - \mathbf{\Theta}^{(t-1)})\|_{\mathrm{F}}^2}\right\}. \qquad (15)$$

The step size strategy ensures decrease in the objective value between successive iterations. This is essential for establishing convergence of the proposed method [cf. Theorem 2]. We remark that the arithmetics in the MCGD method are not affected when we restrict the update of $\mathbf{\Theta}^{(t)}$ in (12) to the entries in $\Xi$ only. This is due to $\mathcal{L}(\boldsymbol{X}) = \mathcal{L}(\mathcal{P}_\Omega(\boldsymbol{X}))$ and the CG update direction (13) only involves the gradient of $\nabla_{\mathbf{\Theta}}\mathcal{L}(\boldsymbol{\alpha}^{(t)}, \mathbf{\Theta}^{(t-1)})$ *w.r.t.* entries of $\mathbf{\

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

# A  Statistical guarantees

## A.1  Main result

We recall the convergence rates for the Frobenius norm of the errors $\Delta\boldsymbol{\Theta} = \hat{\boldsymbol{\Theta}} - \boldsymbol{\Theta}^0$ and $\Delta\boldsymbol{\alpha} = \hat{\boldsymbol{\alpha}} - \boldsymbol{\alpha}^0$ given in Section 2. Define $d_{\mathbf{X}} = \max_k \|\mathbf{X}(k)\|_1$ and the following quantities:

$$\mathsf{D}_{\boldsymbol{\alpha}} = \frac{\|\boldsymbol{\alpha}^0\|_1}{\pi} \frac{\log(n+p)}{\sigma_-^2 \gamma} + \left(\frac{a}{\pi}\right)^2 \log(n+p),$$

$$\mathsf{D}_{\boldsymbol{\Theta}} = \mathsf{D}_{\boldsymbol{\alpha}} + d_{\mathbf{X}} \|\boldsymbol{\alpha}^0\|_1 \left\{ \frac{12\pi\sqrt{\log(n+p)}}{\gamma(1+\nu)a\sigma_+\sqrt{\beta}} + \frac{1}{\pi\sigma_-^2}\left(\frac{\log(n+p)}{\gamma}\right) + 1 \right\}.$$

We assume that $M = (n \vee p)$ is large enough, that is

$$M \geq \left\{ \frac{4\sigma_+^2}{\gamma^6} \log^2\left(\frac{\sqrt{n \wedge p}}{p\gamma\sigma_-}\right) \vee 2\exp\left(\sigma_+^2/\gamma^2 \vee \sigma_+^2\gamma(1+\nu a)\right) \right\}.$$

**Theorem 3** *Assume H1-6. Set*

$$\lambda_L = 2C\sigma_+\sqrt{\pi \max(n,p)\log(n+p)}, \;\; and \;\; \lambda_S \geq 24 \max_k \|\mathbf{X}(k)\|_1 \log(n+p)/\gamma, \qquad (22)$$

*where C is a positive constant. Assume that* $\max(n,p) \geq 4\sigma_+^2/\gamma^6 \log^2(\sqrt{\min(n,p)/(\pi\gamma\sigma_-)}) + 2\exp(\sigma_+^2/\gamma^2 + 2\sigma_+^2\gamma a)$. *Then, with probability at least* $1 - 9(n+p)^{-1}$,

$$
\begin{aligned}
\left\|\hat{\boldsymbol{\alpha}} - \boldsymbol{\alpha}^0\right\|_2^2 &\leq C_1 \frac{sd_{\mathbf{X}}\log(n+p)}{\kappa^2\pi} + \mathsf{D}_{\boldsymbol{\alpha}}, \\
\left\|\hat{\boldsymbol{\Theta}} - \boldsymbol{\Theta}^0\right\|_F^2 &\leq C_2 \left(\frac{r\max(n,p)}{\pi} + \frac{sd_{\mathbf{X}}}{\pi}\right)\log(n+p) + \mathsf{D}_{\boldsymbol{\Theta}}.
\end{aligned}
\qquad (23)
$$

*In* (23), $s := \|\boldsymbol{\alpha}^0\|_0$, $r := \mathrm{rank}(\boldsymbol{\Theta}^0)$. $C_1$ *and* $C_2$ *are positive constants and* $\mathsf{D}_{\boldsymbol{\alpha}}$ *and* $\mathsf{D}_{\boldsymbol{\Theta}}$ *are residuals of lower order whose exact values are given in Appendix A.*

Denoting by $\lesssim$ the inequality up to constant and logarithmic factors, the order of magnitude of the bounds are therefore:

$$
\begin{aligned}
\|\Delta\boldsymbol{\alpha}\|_2^2 &\lesssim \frac{sd_{\mathbf{X}}}{p\kappa^2}, \\
\|\Delta\boldsymbol{\Theta}\|_F^2 &\lesssim \frac{r\beta}{p^2} + \frac{sd_{\mathbf{X}}}{p},
\end{aligned}
$$

where $s = \|\boldsymbol{\alpha}^0\|_0$ and $r = \mathrm{rank}(\boldsymbol{\Theta}^0)$. In the case of almost uniform sampling, *i.e.* $c_1\pi \leq \pi_{ij} \leq c_2\pi$ for all $(i,j) \in [n] \times [p]$ and two positive constants $c_1$ and $c_2$, we obtain that $\beta \leq c_2(n \vee p)\pi$, which yields the following simplified bound:

$$\|\Delta\boldsymbol{\Theta}\|_F^2 \lesssim \frac{rM}{\pi} + \frac{sd_{\mathbf{X}}}{\pi}. \qquad (24)$$

The rate given in (24) is the sum of the usual low-rank convergence rate $rM/p$ and, when $d_{\mathbf{X}}$ is a constant, of the usual sparse vector convergence rate.

## A.2 Sketch of the proof

Let $\{\epsilon_{ij}\}$ be an i.i.d. Rademacher sequence independent of $Y$ and $\Omega$. We define

$$\Sigma_R = \sum_{i=1}^{n}\sum_{j=1}^{p}\omega_{ij}\epsilon_{ij}E_{ij}.$$

In Theorem 4 we give a general result under some assumptions on the regularization parameters $\lambda_L$ and $\lambda_S$, which depend on the random matrices $\nabla\mathcal{L}(\mathbf{M}^0)$ and $\Sigma_R$. Then, Lemma 4 and 5 allow us to compute values of $\lambda_L$ and $\lambda_S$ that satisfy the assumptions of Theorem 4 with high probability. Finally we combining these results yield Theorem 3. Define

$$\Psi_{\boldsymbol{\alpha}} = \frac{\|\boldsymbol{\alpha}^0\|_1}{\pi}\left\{\frac{\lambda_S}{\sigma_-^2} + a^2 d_{\mathbf{X}}\mathbb{E}\|\Sigma_R\|_\infty\right\} + \left(\frac{a}{\pi}\right)^2\log(n+p), \qquad (25)$$

$$\Psi_{\boldsymbol{\Theta}} = \frac{r}{\pi^2}\mathbb{E}\|\Sigma_R\|^2 + \frac{\|\boldsymbol{\alpha}\|_1}{\pi}\left\{\frac{\lambda_S}{(1+\nu)a\lambda_L} + d_{\mathbf{X}}\mathbb{E}\|\Sigma_R\|_\infty\right\} + \Psi_{\boldsymbol{\alpha}}. \qquad (26)$$

**Theorem 4** *Let*

$$\lambda_L \geq 2\left\|\nabla\mathcal{L}(\mathbf{M}^0)\right\|, \quad \lambda_S \geq 2d_{\mathbf{X}}\left(\left\|\nabla\mathcal{L}(\mathbf{M}^0)\right\|_\infty + 2\sigma_+^2(1+\nu)a\right),$$

*and assumptions **H** 2-6 hold. Then, with probability at least* $1 - 8(n+p)^{-1}$

$$
\begin{aligned}
(i) &\qquad \|\Delta\boldsymbol{\alpha}\|_2^2 \leq \frac{C}{\kappa^2}\Psi_{\boldsymbol{\alpha}}, \; and \\
(ii) &\qquad \|\Delta\boldsymbol{\Theta}\|_F^2 \leq C\left\{\frac{r\lambda_L^2}{\pi^2\sigma_-^4} + (1+\nu)a\Psi_{\boldsymbol{\Theta}}\right\}.
\end{aligned}
\qquad (27)
$$

Denote $\Delta\mathbf{M} = \hat{\mathbf{M}} - \mathbf{M}^0$. We first derive an upper bound on the Frobenius error restricted to the observed entries $\|\mathcal{P}_\Omega(\Delta\mathbf{M})\|_F^2$. Then we show some restricted strong convexity property, meaning that $\mathbb{E}\|\mathcal{P}_\Omega(\Delta\mathbf{M})\|_F^2$ is upper bounded by $\|\mathcal{P}_\Omega(\Delta\mathbf{M})\|_F^2$ up to a residual term defined later.

**Upper bound on $\|\mathcal{P}_\Omega(\Delta\mathbf{M})\|_F^2$.** By definition of $\hat{\boldsymbol{\Theta}}$ and $\hat{\boldsymbol{\alpha}}$:

$$\mathcal{L}(\hat{\mathbf{M}}) - \mathcal{L}(\mathbf{M}^0) \leq \lambda_L \left(\left\|\boldsymbol{\Theta}^0\right\|_\star - \left\|\hat{\boldsymbol{\Theta}}\right\|_\star\right) + \lambda_S \left(\left\|\boldsymbol{\alpha}^0\right\|_1 - \|\hat{\boldsymbol{\alpha}}\|_1\right).$$

Recall that, for $\boldsymbol{\alpha} \in \mathbb{R}^q$, we use the notation $f_U(\boldsymbol{\alpha}) = \sum_{k=1}^q \alpha_k \mathbf{X}(k)$. Adding $\langle \nabla\mathcal{L}(\mathbf{M}^0), \Delta\mathbf{M}\rangle$ on both sides of the last inequality, we get

$$\mathcal{L}(\hat{\mathbf{M}}) - \mathcal{L}(\mathbf{M}^0) + \langle\nabla\mathcal{L}(\mathbf{M}^0), \Delta\mathbf{M}\rangle \leq \lambda_L \left(\left\|\boldsymbol{\Theta}^0\right\|_\star - \left\|\hat{\boldsymbol{\Theta}}\right\|_\star\right) - \langle\nabla\mathcal{L}(\mathbf{M}^0), \Delta\boldsymbol{\Theta}\rangle$$
$$+ \lambda_S \left(\left\|\boldsymbol{\alpha}^0\right\|_1 - \|\hat{\boldsymbol{\alpha}}\|_1\right) - \langle\nabla\mathcal{L}(\mathbf{M}^0), f_U(\Delta\boldsymbol{\alpha})\rangle.$$

The strong convexity of the link functions $g_j$, $j \in [p]$, allows us to lower bound the left hand side term and obtain

$$\frac{\sigma_-^2}{2} \|\mathcal{P}_\Omega(\Delta\mathbf{M})\|_F^2 \leq \lambda_L \left(\left\|\boldsymbol{\Theta}^0\right\|_\star - \left\|\hat{\boldsymbol{\Theta}}\right\|_\star\right) - \langle\nabla\mathcal{L}(\mathbf{M}^0), \Delta\boldsymbol{\Theta}\rangle$$
$$+ \lambda_S \left(\left\|\boldsymbol{\alpha}^0\right\|_1 - \|\hat{\boldsymbol{\alpha}}\|_1\right) - \langle\nabla\mathcal{L}(\mathbf{M}^0), f_U(\Delta\boldsymbol{\alpha})\rangle.$$

We now upper bound the right hand side using the following three agruments: the duality of the norms $\|\cdot\|_\star$ and $\|\cdot\|$ on the one hand and of the norms $\|\cdot\|_1$ and $\|\cdot\|_\infty$ on the other hand, the triangular inequality and the following assumptions:

$$\lambda_L \geq 2\left\|\nabla\mathcal{L}(\mathbf{M}^0)\right\|, \quad \lambda_S \geq 2\left\|\nabla\mathcal{L}(\mathbf{M}^0)\right\|_\infty d_{\mathbf{X}}.$$

We obtain

$$\|\mathcal{P}_\Omega(\Delta\mathbf{M})\|_F^2 \leq \frac{3\lambda_L}{\sigma_-^2}\sqrt{2\,\mathrm{rank}(\mathbf{M}^0)} \|\Delta\boldsymbol{\Theta}\|_F + \frac{3\lambda_S}{\sigma_-^2}\left\|\boldsymbol{\alpha}^0\right\|_1. \tag{28}$$

**Restricted strong convexity** We now show that when the errors $\Delta\boldsymbol{\Theta}$ and $\Delta\boldsymbol{\alpha}$ belong to a subspace $\mathcal{C}$ and for a residual D - both defined later on - the following holds with high probability:

$$\|\mathcal{P}_\Omega(\Delta\mathbf{M})\|_F^2 \geq \mathbb{E}\|\mathcal{P}_\Omega(\Delta\mathbf{M})\|_F^2 - \mathsf{D}. \tag{29}$$

We start by defining the set $\mathcal{C}$ and prove that it contains the errors $\Delta\boldsymbol{\Theta}$ and $\Delta\boldsymbol{\alpha}$ with high probability (Lemma 1-2); then we show that restricted strong convexity holds on this subspace (Lemma 3).

For non-negative constants $d_1$, $d_\Pi$, $\rho < m$ and $\varepsilon$ that will be specified later on, define the two following sets:

$$\mathcal{A}(d_1, d_\Pi) = \left\{\boldsymbol{\alpha} \in \mathbb{R}^q : \|\boldsymbol{\alpha}\|_1 \leq d_1, \|\mathcal{P}_\Omega(f_U(\boldsymbol{\alpha}))\|_F^2 \leq d_\Pi\right\}. \tag{30}$$

The constants $d_1$ and $d_\Pi$ define the constraints on the $\ell_1$ norm of $\boldsymbol{\alpha}$ and weighted Frobenius norm of $f_U(\boldsymbol{\alpha})$.

$$\mathcal{L}(\rho, \varepsilon) = \left\{\boldsymbol{\Theta} \in \mathbb{R}^{n\times p}, \boldsymbol{\alpha} \in \mathbb{R}^q : \|\mathcal{P}_\Omega(\boldsymbol{\Theta} + f_U(\boldsymbol{\alpha}))\|_F^2 \geq \frac{72\log(n+p)}{\pi\log(6/5)},\right.$$
$$\left.\|\boldsymbol{\Theta} + f_U(\boldsymbol{\alpha})\|_\infty \leq 1, \|\boldsymbol{\Theta}\|_\star \leq \sqrt{\rho}\|\boldsymbol{\Theta}\|_F + \varepsilon\right\} \tag{31}$$

Condition $\|\boldsymbol{\Theta}\|_\star \leq \sqrt{\rho}\|\boldsymbol{\Theta}\|_F + \varepsilon$ is a relaxed form of the condition $\|\boldsymbol{\Theta}\|_\star \leq \sqrt{\rho}\|\boldsymbol{\Theta}\|_F$ satisfied for matrices of rank $\rho$. Finally, we define the constrained set of interest:

$$\mathcal{C}(d_1, d_\Pi, \rho, \varepsilon) = \mathcal{L}(\rho, \varepsilon) \cap \left\{\mathbb{R}^{n\times p} \times \mathcal{A}(d_1, d_\Pi)\right\}.$$

Let

$$d_1 = 4\|\boldsymbol{\alpha}\|_1,$$
$$d_\Pi = \frac{3\lambda_S}{\sigma_-^2}\left\|\boldsymbol{\alpha}^0\right\|_1 + 64a^2 d_{\mathbf{X}}\mathbb{E}\|\Sigma_R\|_\infty\|\boldsymbol{\alpha}\|_1 + 3072a^2\pi^{-1} + \frac{72a^2\log(n+p)}{\log(6/5)}.$$

The following Lemma, proved in Appendix B.1 states that with high probability, $\Delta\boldsymbol{\alpha} \in \mathcal{A}(d_1, d_\Pi)$.

**Lemma 1** *Let* $\lambda_S \geq 2d_{\mathbf{X}} \left( \left\| \nabla \mathcal{L}(\mathbf{M}^0) \right\|_\infty + 2\sigma_+^2 (1 + d_{\mathbf{X}})a \right)$ *and assume **H** 2-6 hold. Then, with probability at least* $1 - 8(n+p)^{-1}$,

$$\Delta\boldsymbol{\alpha} \in \mathcal{A}(d_1, d_\Pi);$$

Lemma 1 (proved in Appendix B.2) implies $(i)$ of Theorem 4. Thus, we only need to prove $(ii)$.

**Lemma 2** *Let*

$$\lambda_L \geq 2 \left\| \nabla \mathcal{L}(\mathbf{M}^0) \right\|, \quad \lambda_S \geq 2d_{\mathbf{X}} \left( \left\| \nabla \mathcal{L}(\mathbf{M}^0) \right\|_\infty + 2\sigma_+^2 (1 + d_{\mathbf{X}})a \right),$$

*and assumption **H** 4 hold. Then, for* $\rho = 32r$ *and* $\varepsilon = 3\lambda_S / \lambda_L \left\| \boldsymbol{\alpha}^0 \right\|_1$,

$$\left\| \Delta\boldsymbol{\Theta} \right\|_\star \leq \sqrt{\rho} \left\| \Delta\boldsymbol{\Theta} \right\|_F + \varepsilon.$$

A proof of Lemma 2 can be found in Appendix B.2. As a consequence, under the conditions on the regularization parameters $\lambda_L$ and $\lambda_S$ given in Lemma 2 and whenever

$$\mathbb{E} \left\| \mathcal{P}_\Omega(\Delta\boldsymbol{\Theta} + f_U(\Delta\boldsymbol{\alpha})) \right\|_F^2 \geq \frac{72 \log(n+p)}{\pi \log(6/5)},$$

the error terms $(\Delta\boldsymbol{\Theta}, \Delta\boldsymbol{\alpha})$ belong to the constrained set $\mathcal{C}(d_1, d_\Pi, \rho, \varepsilon)$ with high probability. We therefore consider the two possible cases: $\mathbb{E} \left\| \mathcal{P}_\Omega(\Delta\boldsymbol{\Theta} + f_U(\Delta\boldsymbol{\alpha})) \right\|_F^2 < \frac{72 \log(n+p)}{\pi \log(6/5)}$ and $\mathbb{E} \left\| \mathcal{P}_\Omega(\Delta\boldsymbol{\Theta} + f_U(\Delta\boldsymbol{\alpha})) \right\|_F^2 \geq \frac{72 \log(n+p)}{\pi \log(6/5)}$.

**Case 1:** Suppose $\mathbb{E} \left\| \mathcal{P}_\Omega(\Delta\boldsymbol{\Theta} + f_U(\Delta\boldsymbol{\alpha})) \right\|_F^2 < \frac{72 \log(n+p)}{\pi \log(6/5)}$. Then, Lemma 1 combined with the fact that $\left\| \mathbf{M} \right\|_F^2 \leq \pi^{-1} \left\| \mathcal{P}_\Omega(\mathbf{M}) \right\|_F^2$ for all $\mathbf{M}$, and the identity $(a+b)^2 \geq a^2/4 - 4b^2$ ensures that

$$\left\| \Delta\boldsymbol{\Theta} \right\|_F^2 \leq 4 \left\| \Delta\boldsymbol{\Theta} + f_U(\Delta\boldsymbol{\alpha}) \right\|_F^2 + 16 \left\| f_U(\Delta\boldsymbol{\alpha}) \right\|_F^2,$$

therefore

$$\left\| \Delta\boldsymbol{\Theta} \right\|_F^2 \leq \frac{288a^2 \log(n+p)}{\log(6/5)} + 16\Phi_{\boldsymbol{\alpha}},$$

which implies (ii) of Theorem 4.

**Case 2:** Suppose $\mathbb{E} \left\| \mathcal{P}_\Omega(\Delta\boldsymbol{\Theta} + f_U(\Delta\boldsymbol{\alpha})) \right\|_F^2 \geq \frac{72 \log(n+p)}{\pi \log(6/5)}$. Then, Lemma 1 and 2 yield that with probability at least $1 - 8(n+p)^{-1}$,

$$\left( \frac{\Delta\boldsymbol{\Theta}}{2(1+\nu)a}, \frac{\Delta\boldsymbol{\alpha}}{2(1+\nu)a} \right) \in \mathcal{C}(d_1', d_\Pi', \rho', \varepsilon'), \text{ with}$$

$$\begin{aligned} d_1' &= \frac{d_1}{2(1+\nu)a}, & d_\Pi' &= \frac{d_\Pi}{4(1+\nu)^2 a^2}, \\ \rho' &= \rho, & \varepsilon' &= \frac{\varepsilon}{2(1+\nu)a}, \end{aligned}$$

where $d_1, d_\Pi, \rho$ and $\varepsilon$ are defined in Lemma 1 and 2. We use the following result, proved in Appendix B.3. Define the set $\tilde{\mathcal{A}}(d_1)$ as follows:

$$\tilde{\mathcal{A}}(d_1) = \left\{ \boldsymbol{\alpha} \in \mathbb{R}^q : \quad \left\| \boldsymbol{\alpha} \right\|_\infty \leq 1; \quad \left\| \boldsymbol{\alpha} \right\|_1 \leq d_1; \quad \left\| \mathcal{P}_\Omega(f_U\boldsymbol{\alpha}) \right\|_F^2 \geq \frac{18 \log(n+p)}{\pi \log(6/5)} \right\}.$$

Let $d_1, d_\Pi, \rho$ and $\varepsilon$ be positive constants, and

$$\begin{aligned} \mathsf{D}_{\boldsymbol{\alpha}} &= 8\nu d_1 d_{\mathbf{X}} \mathbb{E} \|\Sigma_R\|_\infty + 768\pi^{-1}, \\ \mathsf{D}_X &= \frac{112\rho}{\pi} \mathbb{E} \|\Sigma_R\|^2 + 8\nu\varepsilon \mathbb{E}\|\Sigma_R\| + 8\nu d_1 d_{\mathbf{X}} \mathbb{E}\|\Sigma_R\|_\infty + d_\Pi + 768\pi^{-1}. \end{aligned} \tag{32}$$

**Lemma 3** *Assume **H** 6. Then, the following properties hold:*

*(i) For any $\boldsymbol{\alpha} \in \tilde{\mathcal{A}}(d_1)$, with probability at least $1 - 8(n+p)^{-1}$,*

$$\|\mathcal{P}_\Omega(f_U(\boldsymbol{\alpha}))\|_F^2 \geq \frac{1}{2}\mathbb{E}\|\mathcal{P}_\Omega(f_U(\boldsymbol{\alpha}))\|_F^2 - \mathsf{D}_{\boldsymbol{\alpha}}.$$

*(ii) For any pair $(\boldsymbol{\Theta}, \boldsymbol{\alpha}) \in \mathcal{C}(d_1, d_\Pi, \rho, \varepsilon)$, with probability at least $1 - 8(n+p)^{-1}$*

$$\|\mathcal{P}_\Omega(\Delta\boldsymbol{\Theta} + f_U(\Delta\boldsymbol{\alpha}))\|_F^2 \geq \frac{1}{2}\mathbb{E}\|\mathcal{P}_\Omega(\Delta\boldsymbol{\Theta} + f_U(\Delta\boldsymbol{\alpha}))\|_F^2 - \mathsf{D}_X. \tag{33}$$

Lemma 3 is proved in Appendix B.3. We apply Lemma 3 (ii) to $\left(\frac{\Delta\boldsymbol{\Theta}}{2(1+\nu)a}, \frac{\Delta\boldsymbol{\alpha}}{2(1+\nu)a}\right)$ which implies that with probability at least $1 - 8(n+p)^{-1}$, $\mathbb{E}\mathbb{E}\|\mathcal{P}_\Omega(\Delta\mathbf{M})\|_F^2 \leq 2\|\mathcal{P}_\Omega(\Delta\mathbf{M})\|_F^2 + 2(1+\nu)a\Psi_{\boldsymbol{\Theta}}$. Combined with (28) and $\|\Delta\mathbf{M}\|_F^2 \leq \pi^{-1}\mathbb{E}\|\mathcal{P}_\Omega(\Delta\mathbf{M})\|_F^2$, it implies that

$$\|\Delta\mathbf{M}\|_F^2 \leq \frac{6\sqrt{2r}\lambda_L}{p\sigma_-^2}\|\Delta\boldsymbol{\Theta}\|_F + \frac{6\lambda_S}{\pi\sigma_-^2}\|\boldsymbol{\alpha}^0\|_1 + 2(1+\nu)a\Psi_{\boldsymbol{\Theta}}.$$

Now using $\|\Delta\mathbf{M}\|_F^2 \geq \frac{\|\Delta\boldsymbol{\Theta}\|_F^2}{2} - \|f_U(\Delta\boldsymbol{\alpha})\|_F^2$ and $\frac{6\sqrt{2r}\lambda_L}{\pi\sigma_-^2}\|\Delta\boldsymbol{\Theta}\|_F \leq \frac{\|\Delta\boldsymbol{\Theta}\|_F^2}{4} + \frac{288r\lambda_L^2}{p^2\sigma_-^4}$, we obtain

$$\|\Delta\boldsymbol{\Theta}\|_F^2 \leq \frac{1152r\lambda_L^2}{p^2\sigma_-^4} + \frac{24\lambda_S\|\boldsymbol{\alpha}^0\|_1}{\pi\sigma_-^2} + 2(1+\nu)a\Psi_{\boldsymbol{\Theta}} + 4\Psi_{\boldsymbol{\alpha}},$$

which gives the result of Theorem 4 (ii).

We now give deterministic upper bounds on $\mathbb{E}\|\Sigma_R\|$ and $\mathbb{E}\|\Sigma_R\|_\infty$, and probabilistic upper bounds on $\|\nabla\mathcal{L}(\mathbf{M}^0)\|$ and $\|\nabla\mathcal{L}(\mathbf{M}^0)\|_\infty$. We will use them to select values of $\lambda_L$ and $\lambda_S$ which satisfy the assumptions of Theorem 4 and compute the corresponding upper bounds.

**Lemma 4** *[21, Lemma 10] Let assumption **H** 6 hold. Then, there exists an absolute constant $C^\star$ such that the two following inequalities hold*

$$\mathbb{E}\|\Sigma_R\|_\infty \leq 1, \text{ and}$$

$$\mathbb{E}\|\Sigma_R\| \leq C^\star\left\{\sqrt{\beta} + \sqrt{\log(\min(n,p))}\right\}.$$

**Lemma 5** *[21, Lemma 10] Let assumptions **H** 1-6 hold. Then, there exists an absolute constant $c^\star$ such that the following two inequalities hold with probability at least $1 - (n+p)^{-1}$.*

$$\|\nabla\mathcal{L}(\mathbf{M}^0)\|_\infty \leq 6\max\left\{\sigma_+\sqrt{\log(n+p)}, \frac{\log(n+p)}{\gamma}\right\}, \tag{34}$$

$$\|\nabla\mathcal{L}(\mathbf{M}^0)\| \leq c^\star\max\left\{\sigma_+\sqrt{\beta\log(n+p)}, \frac{\log(n+p)}{\gamma}\log\left(\frac{1}{\sigma_-}\sqrt{\frac{np}{\beta}}\right)\right\}. \tag{35}$$

From Theorem 4, Lemma 4 and 5 combined with a union bound argument, we deduce result given in Section 2.

# B  Technical results

## B.1  Proof of Lemma 1

We start by proving $\|\Delta\boldsymbol{\alpha}\|_1 \leq 4\|\boldsymbol{\alpha}^0\|_1$. By the optimality conditions over a convex set [2, Chapter 4, Section 2, Proposition 4], there exist two subgradients $\hat{f}_{\boldsymbol{\Theta}}$ in the subdifferential of $\|\cdot\|_\star$ taken at $\hat{\boldsymbol{\Theta}}$ and $\hat{f}_{\boldsymbol{\alpha}}$ in the subdifferential of $\|\cdot\|_1$ taken at $\hat{\boldsymbol{\alpha}}$, such that for all feasible pairs $(\boldsymbol{\Theta}, \boldsymbol{\alpha})$ we have

$$\langle\nabla\mathcal{L}(\hat{\mathbf{M}}), \boldsymbol{\Theta} - \hat{\boldsymbol{\Theta}} + \sum_{k=1}^q(\boldsymbol{\alpha}_k - \hat{\boldsymbol{\alpha}}_k)\mathbf{X}(k)\rangle + \lambda_L\langle\hat{f}_{\boldsymbol{\Theta}}, \boldsymbol{\Theta} - \hat{\boldsymbol{\Theta}}\rangle + \lambda_S\langle\hat{f}_{\boldsymbol{\alpha}}, \boldsymbol{\alpha} - \hat{\boldsymbol{\alpha}}\rangle \geq 0. \tag{36}$$

Applying inequality (36) to the pair $(\hat{\boldsymbol{\Theta}}, \boldsymbol{\alpha}^0)$ we obtain

$$\langle \nabla \mathcal{L}(\hat{\mathbf{M}}), \sum_{k=1}^{q} \Delta \boldsymbol{\alpha}_k \mathbf{X}(k) \rangle + \lambda_S \langle \hat{f}_{\boldsymbol{\alpha}}, \Delta \boldsymbol{\alpha} \rangle \geq 0.$$

Denote $\tilde{\mathbf{M}} = \hat{\boldsymbol{\Theta}} + \sum_{k=1}^{q} \boldsymbol{\alpha}_k^0 \mathbf{X}(k)$. The last inequality is equivalent to

$$\underbrace{\langle \nabla \mathcal{L}(\mathbf{M}^0), f_U(\Delta \boldsymbol{\alpha}) \rangle}_{B_1} + \underbrace{\langle \nabla \mathcal{L}(\tilde{\mathbf{M}}) - \nabla \mathcal{L}(\mathbf{M}^0), f_U(\Delta \boldsymbol{\alpha}) \rangle}_{B_2} + \underbrace{\langle \nabla \mathcal{L}(\hat{\mathbf{M}}) - \nabla \mathcal{L}(\tilde{\mathbf{M}}), f_U(\Delta \boldsymbol{\alpha}) \rangle}_{B_3}$$
$$+ \lambda_S \langle \hat{f}_{\boldsymbol{\alpha}}, \Delta \boldsymbol{\alpha} \rangle \geq 0.$$

We now derive upper bounds on the three terms $B_1$, $B_2$ and $B_3$ separately. Recall that we denote $d_{\mathbf{X}} = \max_k \|\mathbf{X}(k)\|_1$ and bound $B_1$ as follows:

$$B_1 \leq \|\Delta \boldsymbol{\alpha}\|_1 \|\nabla \mathcal{L}(\mathbf{M}^0)\|_\infty d_{\mathbf{X}}. \tag{37}$$

Similarly, the duality between $\|\cdot\|_\infty$ and $\|\cdot\|_1$ gives

$$B_2 \leq \|\Delta \boldsymbol{\alpha}\|_1 \left\|\nabla \mathcal{L}(\tilde{\mathbf{M}}) - \nabla \mathcal{L}(\mathbf{M}^0)\right\|_\infty d_{\mathbf{X}}.$$

Moreover, $\nabla \mathcal{L}(\tilde{\mathbf{M}}) - \nabla \mathcal{L}(\mathbf{M}^0)$ is a matrix with entries $g_j'(\tilde{\mathbf{M}}_{ij}) - g_j'(\mathbf{M}_{ij}^0)$, therefore assumption **H** 4 ensures

$$\left\|\nabla \mathcal{L}(\tilde{\mathbf{M}}) - \nabla \mathcal{L}(\mathbf{M}^0)\right\|_\infty \leq 2\sigma_+^2(1+\nu)a,$$

and finally we obtain

$$B_2 \leq \|\Delta \boldsymbol{\alpha}\|_1 2\sigma_+^2(1+\nu)a d_{\mathbf{X}}. \tag{38}$$

We finally bound $B_3$ as follows. We have that

$$B_3 = \sum_{i=1}^{n} \sum_{j=1}^{p} \omega_{ij} \left( g_j'(\hat{\mathbf{M}}_{ij}) - g_j'(\tilde{\mathbf{M}}_{ij}) \right) \left( \tilde{\mathbf{M}}_{ij} - \hat{\mathbf{M}}_{ij} \right).$$

Now, for all $j \in [p]$, $g_j'$ is increasing therefore

$$\left( g_j'(\hat{\mathbf{M}}_{ij}) - g_j'(\tilde{\mathbf{M}}_{ij}) \right) \left( \tilde{\mathbf{M}}_{ij} - \hat{\mathbf{M}}_{ij} \right) \leq 0,$$

which implies $B_3 \leq 0$. Combined with (37) and (38) this yields

$$\lambda_S \langle \hat{f}_{\boldsymbol{\alpha}}, \hat{\boldsymbol{\alpha}} - \boldsymbol{\alpha} \rangle \leq \|\Delta \boldsymbol{\alpha}\|_1 d_{\mathbf{X}} \left( \|\nabla \mathcal{L}(\mathbf{M}^0)\|_\infty + 2\sigma_+^2(1+\nu)a \right).$$

Besides, the convexity of $\|\cdot\|_1$ gives $\langle \hat{f}_{\boldsymbol{\alpha}}, \hat{\boldsymbol{\alpha}} - \boldsymbol{\alpha} \rangle \geq \|\hat{\boldsymbol{\alpha}}\|_1 - \|\boldsymbol{\alpha}\|_1$, therefore

$$\left\{ \lambda_S - d_{\mathbf{X}} \left( \|\nabla \mathcal{L}(\mathbf{M}^0)\|_\infty + 2\sigma_+^2(1+\nu)a \right) \right\} \|\hat{\boldsymbol{\alpha}}\|_1 \leq$$
$$\left\{ \lambda_S + d_{\mathbf{X}} \left( \|\nabla \mathcal{L}(\mathbf{M}^0)\|_\infty + 2\sigma_+^2(1+\nu)a \right) \right\} \|\boldsymbol{\alpha}\|_1,$$

and the condition $\lambda_S \geq 2 \left\{ d_{\mathbf{X}} \left( \|\nabla \mathcal{L}(\mathbf{M}^0)\|_\infty + 2\sigma_+^2(1+\nu)a \right) \right\}$ gives $\|\hat{\boldsymbol{\alpha}}\|_1 \leq 3 \|\boldsymbol{\alpha}\|_1$ and finally

$$\|\Delta \boldsymbol{\alpha}\|_1 \leq 4 \|\boldsymbol{\alpha}\|_1. \tag{39}$$

We consider the two following cases.

**Case 1:** $\mathbb{E} \|\mathcal{P}_\Omega(f_U(\Delta \boldsymbol{\alpha}))\|_F^2 < \frac{72a^2 \log(n+p)}{\pi \log(6/5)}$. Then the result holds trivially.

**Case 2:** $\mathbb{E} \left\| \mathcal{P}_\Omega(f_U(\Delta\boldsymbol{\alpha})) \right\|_F^2 \geq \frac{72a^2 \log(n+p)}{\pi \log(6/5)}$. For $d_1 > 0$ recall the definition of the set

$$\tilde{\mathcal{A}}(d_1) = \left\{ \boldsymbol{\alpha} \in \mathbb{R}^q : \quad \left\| \boldsymbol{\alpha} \right\|_\infty \leq 1; \quad \left\| \boldsymbol{\alpha} \right\|_1 \leq d_1; \quad \mathbb{E} \left\| \mathcal{P}_\Omega(f_U(\Delta\boldsymbol{\alpha})) \right\|_F^2 \geq \frac{18 \log(n+p)}{\pi \log(6/5)} \right\}.$$

Inequality (39) and $\left\| \Delta\boldsymbol{\alpha} \right\|_\infty \leq 2a$ imply that

$$\frac{\Delta\boldsymbol{\alpha}}{2a} \in \tilde{\mathcal{A}} \left( \frac{2 \left\| \boldsymbol{\alpha} \right\|_1}{a} \right).$$

Therefore we can apply Lemma 3(i) and obtain that with probability at least $1 - 8(n+p)^{-1}$,

$$\mathbb{E} \left\| \mathcal{P}_\Omega(f_U(\Delta\boldsymbol{\alpha})) \right\|^2 \leq 2 \left\| \mathcal{P}_\Omega(f_U(\Delta\boldsymbol{\alpha})) \right\|_F^2 + 64\nu a \left\| \boldsymbol{\alpha} \right\|_1 d_{\mathbf{X}} \mathbb{E} \left[ \left\| \Sigma_R \right\|_\infty \right] + 3072 a^2 p^{-1}. \quad (40)$$

We now must upper bound the quantity $\left\| \mathcal{P}_\Omega(f_U(\Delta\boldsymbol{\alpha})) \right\|_F^2$. Recall that $\tilde{\mathbf{M}} = \sum_{k=1}^q \boldsymbol{\alpha}_k \mathbf{X}(k) + \hat{\mathbf{M}}$. By definition,

$$\mathcal{L}(\hat{\mathbf{X}}) + \lambda_L \left\| \hat{\boldsymbol{\Theta}} \right\|_\star + \lambda_S \left\| \hat{\boldsymbol{\alpha}} \right\|_1 \leq \mathcal{L}(\tilde{\mathbf{M}}) + \lambda_L \left\| \hat{\boldsymbol{\Theta}} \right\|_\star + \lambda_S \left\| \boldsymbol{\alpha} \right\|_1,$$

i.e.

$$\mathcal{L}(\hat{\mathbf{M}}) - \mathcal{L}(\tilde{\mathbf{M}}) \leq \lambda_S \left( \left\| \boldsymbol{\alpha} \right\|_1 - \left\| \hat{\boldsymbol{\alpha}} \right\|_1 \right).$$

Substracting $\langle \nabla\mathcal{L}(\tilde{\mathbf{M}}), \hat{\mathbf{M}} - \tilde{\mathbf{M}} \rangle$ on both sides and by strong convexity of $\mathcal{L}$ we obtain

$$\begin{aligned}
\frac{\sigma_-^2}{2} \left\| \mathcal{P}_\Omega(f_U(\Delta\boldsymbol{\alpha})) \right\|^2 \quad & \leq \lambda_S \left( \left\| \boldsymbol{\alpha} \right\|_1 - \left\| \hat{\boldsymbol{\alpha}} \right\|_1 \right) + \langle \nabla\mathcal{L}(\tilde{\mathbf{M}}), f_U(\Delta\boldsymbol{\alpha}) \rangle \\
& \leq \lambda_S \left( \left\| \boldsymbol{\alpha} \right\|_1 - \left\| \hat{\boldsymbol{\alpha}} \right\|_1 \right) + \underbrace{\left| \langle \nabla\mathcal{L}(\mathbf{M}^0), f_U(\Delta\boldsymbol{\alpha}) \rangle \right|}_{\mathsf{C}_1}. \\
& + \underbrace{\left| \langle \nabla\mathcal{L}(\mathbf{M}^0) - \nabla\mathcal{L}(\tilde{\mathbf{M}}), f_U(\Delta\boldsymbol{\alpha}) \rangle \right|}_{\mathsf{C}_2}
\end{aligned} \quad (41)$$

The duality of $\left\| \cdot \right\|_1$ and $\left\| \cdot \right\|_\infty$ yields $\mathsf{C}_1 \leq \left\| \nabla\mathcal{L}(\mathbf{M}^0) \right\|_\infty d_{\mathbf{X}} \left\| \Delta\boldsymbol{\alpha} \right\|_1$, and

$$\mathsf{C}_2 \leq \left\| \nabla\mathcal{L}(\mathbf{M}^0) - \nabla\mathcal{L}(\tilde{\mathbf{M}}) \right\|_\infty d_{\mathbf{X}} \left\| \Delta\boldsymbol{\alpha} \right\|_1.$$

Furthermore,

$$\left\| \nabla\mathcal{L}(\mathbf{M}^0) - \nabla\mathcal{L}(\tilde{\mathbf{M}}) \right\|_\infty \leq 2\sigma_+^2 a,$$

since for all $(i,j) \in [n] \times [p]$ $|\tilde{\mathbf{M}}_{ij} - \mathbf{M}_{ij}^0| \leq 2a$ and $g_j''(\tilde{\mathbf{M}}_{ij}) \leq \sigma_+^2$. The last three inequalities plugged in (41) give

$$\frac{\sigma_-^2}{2} \left\| \mathcal{P}_\Omega(f_U(\Delta\boldsymbol{\alpha})) \right\|_F^2 \quad \leq \lambda_S \left( \left\| \boldsymbol{\alpha} \right\|_1 - \left\| \hat{\boldsymbol{\alpha}} \right\|_1 \right) + d_{\mathbf{X}} \left\| \Delta\boldsymbol{\alpha} \right\|_1 \left\{ \left\| \nabla\mathcal{L}(\mathbf{M}^0) \right\|_\infty + 2\sigma_+^2 a \right\}.$$

The triangular inequality gives

$$\begin{aligned}
\frac{\sigma_-^2}{2} \left\| \mathcal{P}_\Omega(f_U(\Delta\boldsymbol{\alpha})) \right\|_F^2 \quad & \leq \left\{ d_{\mathbf{X}} \left( \left\| \nabla\mathcal{L}(\mathbf{M}^0) \right\|_\infty + 2\sigma_+^2 a \right) + \lambda_S \right\} \left\| \boldsymbol{\alpha} \right\|_1 \\
& + \left\{ d_{\mathbf{X}} \left( \left\| \nabla\mathcal{L}(\mathbf{M}^0) \right\|_\infty + 2\sigma_+^2 a \right) - \lambda_S \right\} \left\| \hat{\boldsymbol{\alpha}} \right\|_1.
\end{aligned}$$

Then, the assumption $\lambda_S \geq 2d_{\mathbf{X}} \left( \left\| \nabla\mathcal{L}(\mathbf{M}^0) \right\|_\infty + 2\sigma_+^2(1+\nu)a \right)$ gives

$$\left\| \mathcal{P}_\Omega(f_U(\Delta\boldsymbol{\alpha})) \right\|_F^2 \leq \frac{3\lambda_S}{\sigma_-^2} \left\| \boldsymbol{\alpha} \right\|_1.$$

Plugged into (40), this last inequality implies that with probability at least $1 - 8(n+p)^{-1}$

$$\mathbb{E} \left\| \mathcal{P}_\Omega(f_U(\Delta\boldsymbol{\alpha})) \right\|_F^2 \leq \frac{3\lambda_S}{\sigma_-^2} \left\| \boldsymbol{\alpha} \right\|_1 + 64\nu a \left\| \boldsymbol{\alpha} \right\|_1 d_{\mathbf{X}} \mathbb{E} \left[ \left\| \Sigma_R \right\|_\infty \right] + 3072 a^2 p^{-1}. \quad (42)$$

Combining (39) and (42) gives the result.

## B.2 Proof of Lemma 2

Using (36) for $L = \boldsymbol{\Theta}^0$ and $\boldsymbol{\alpha} = \boldsymbol{\alpha}$ we obtain

$$\langle \nabla \mathcal{L}(\hat{\mathbf{M}}), \Delta \boldsymbol{\Theta} + \sum_{k=1}^{q} (\Delta \boldsymbol{\alpha}_k) \mathbf{X}(k) \rangle + \lambda_L \langle \hat{f}_L, \Delta \boldsymbol{\Theta} \rangle + \lambda_S \langle \hat{f}_{\boldsymbol{\alpha}}, \Delta \boldsymbol{\alpha} \rangle \geq 0.$$

Then, the convexity of $\|\cdot\|_*$ and $\|\cdot\|_1$ imply that

$$\left\| \boldsymbol{\Theta}^0 \right\|_\star \geq \left\| \hat{\boldsymbol{\Theta}} \right\|_\star + \langle \partial \left\| \hat{\boldsymbol{\Theta}} \right\|_\star, \Delta \boldsymbol{\Theta} \rangle,$$
$$\|\boldsymbol{\alpha}\|_1 \geq \|\hat{\boldsymbol{\alpha}}\|_\star + \langle \partial \|\hat{\boldsymbol{\alpha}}\|_1, \Delta \boldsymbol{\alpha} \rangle.$$

The last three inequalities yield

$$\lambda_L \left( \left\| \hat{\boldsymbol{\Theta}} \right\|_\star - \left\| \boldsymbol{\Theta}^0 \right\|_\star \right) + \lambda_S (\|\hat{\boldsymbol{\alpha}}\|_1 - \|\boldsymbol{\alpha}\|_1) \leq \langle \nabla \mathcal{L}(\hat{\mathbf{M}}), \Delta \boldsymbol{\Theta} \rangle$$
$$+ \langle \nabla \mathcal{L}(\hat{\mathbf{M}}), \sum_{k=1}^{q} (\Delta \boldsymbol{\alpha}_k) \mathbf{X}(k) \rangle$$
$$\leq \left\| \nabla \mathcal{L}(\hat{\mathbf{M}}) \right\| \|\Delta \boldsymbol{\Theta}\|_\star + d_{\mathbf{X}} \left\| \nabla \mathcal{L}(\hat{\mathbf{M}}) \right\|_\infty \|\Delta \boldsymbol{\alpha}\|_1.$$

Using the conditions

$$\lambda_L \geq 2 \left\| \nabla \mathcal{L}(\mathbf{M}^0) \right\|, \quad \lambda_S \geq 2 d_{\mathbf{X}} \left\{ \left\| \nabla \mathcal{L}(\mathbf{M}^0) \right\|_\infty + 2\sigma_+^2 (1+\nu) a \right\},$$

we get

$$\lambda_L \left( \left\| P_{\boldsymbol{\Theta}^0}^\perp (\Delta \boldsymbol{\Theta}) \right\|_\star - \| P_{\boldsymbol{\Theta}^0}(\Delta \boldsymbol{\Theta}) \|_\star \right) + \lambda_S (\|\hat{\boldsymbol{\alpha}}\|_1 - \|\boldsymbol{\alpha}\|_1) \leq$$
$$\frac{\lambda_L}{2} \left( \left\| P_{\boldsymbol{\Theta}^0}^\perp (\Delta \boldsymbol{\Theta}) \right\|_\star + \| P_{\boldsymbol{\Theta}^0}(\Delta \boldsymbol{\Theta}) \|_\star \right) + \frac{\lambda_S}{2} \|\Delta \boldsymbol{\alpha}\|_1,$$

which implies

$$\left\| P_{\boldsymbol{\Theta}^0}^\perp (\Delta \boldsymbol{\Theta}) \right\|_\star \leq 3 \| P_{\boldsymbol{\Theta}^0}(\Delta \boldsymbol{\Theta}) \|_\star + 3\lambda_S / \lambda_L \|\boldsymbol{\alpha}\|_1.$$

Now, using

$$\|\Delta \boldsymbol{\Theta}\|_\star \leq \left\| P_{\boldsymbol{\Theta}^0}^\perp (\Delta \boldsymbol{\Theta}) \right\|_\star + \| P_{\boldsymbol{\Theta}^0}(\Delta \boldsymbol{\Theta}) \|_\star, \quad \| P_{\boldsymbol{\Theta}^0}(\Delta \boldsymbol{\Theta}) \|_F \leq \|\Delta \boldsymbol{\Theta}\|_F$$

and $\text{rank}(P_{\boldsymbol{\Theta}^0}(\Delta \boldsymbol{\Theta})) \leq 2r$, we get

$$\|\Delta \boldsymbol{\Theta}\|_\star \leq \sqrt{32r} \|\Delta \boldsymbol{\Theta}\|_F + 3\lambda_S / \lambda_L \|\boldsymbol{\alpha}\|_1.$$

This completes the proof of Lemma 2.

## B.3 Proof of Lemma 3

**Proof of (i):**  Recall
$$\mathsf{D}_{\boldsymbol{\alpha}} = 8\nu d_1 d_{\mathbf{X}} \mathbb{E} \left[ \|\Sigma_R\|_\infty \right] + 768 p^{-1}$$

and

$$\tilde{\mathcal{A}}(d_1) = \left\{ \boldsymbol{\alpha} \in \mathbb{R}^q : \quad \|\boldsymbol{\alpha}\|_\infty \leq 1; \quad \|\boldsymbol{\alpha}\|_1 \leq d_1; \quad \mathbb{E} \|\mathcal{P}_\Omega(\mathsf{f}_U(\boldsymbol{\alpha}))\|_F^2 \geq \frac{18 \log(n+p)}{\pi \log(6/5)} \right\}.$$

We will show that the probability of the following event is small:

$$\mathcal{B} = \left\{ \exists \boldsymbol{\alpha} \in \tilde{\mathcal{A}}(d_1) \text{ such that } \left| \|\mathcal{P}_\Omega(\mathsf{f}_U(\boldsymbol{\alpha}))\|_F^2 - \mathbb{E} \|\mathcal{P}_\Omega(\mathsf{f}_U(\boldsymbol{\alpha}))\|_F^2 \right| > \frac{1}{2} \mathbb{E} \|\mathcal{P}_\Omega(\mathsf{f}_U(\boldsymbol{\alpha}))\|_F^2 + \mathsf{D}_{\boldsymbol{\alpha}} \right\}.$$

Indeed, $\mathcal{B}$ contains the complement of the event we are interested in. We use a peeling argument to upper bound the probability of event $\mathcal{B}$. Let $\nu = \frac{18 \log(n+p)}{\pi \log(6/5)}$ and $\eta = 6/5$. For $l \in \mathbb{N}$ set

$$\mathcal{S}_l = \left\{ \boldsymbol{\alpha} \in \tilde{\mathcal{A}}(d_1) : \quad \eta^{l-1} \nu \leq \mathbb{E} \|\mathcal{P}_\Omega(\mathsf{f}_U(\boldsymbol{\alpha}))\|_F^2 \leq \eta^l \nu \right\}.$$

Under the event $\mathcal{B}$, there exists $l \geq 1$ and $\boldsymbol{\alpha} \in \tilde{\mathcal{A}}(d_1) \cap S_l$ such that

$$
\begin{aligned}
\left| \|\mathcal{P}_\Omega(\mathsf{f}_U(\boldsymbol{\alpha}))\|^2 - \mathbb{E}\,\|\mathcal{P}_\Omega(\mathsf{f}_U(\boldsymbol{\alpha}))\|_F^2 \right| \;
&> \frac{1}{2}\mathbb{E}\,\|\mathcal{P}_\Omega(\mathsf{f}_U(\boldsymbol{\alpha}))\|_F^2 + \mathsf{D}_{\boldsymbol{\alpha}} \\
&> \frac{1}{2}\eta^{l-1}\nu + \mathsf{D}_{\boldsymbol{\alpha}} \qquad (43) \\
&= \frac{5}{12}\eta^l \nu + \mathsf{D}_{\boldsymbol{\alpha}}.
\end{aligned}
$$

For $T > \nu$, consider the set of vectors

$$
\tilde{\mathcal{A}}(d_1, T) = \left\{ \boldsymbol{\alpha} \in \tilde{\mathcal{A}}(d_1) : \mathbb{E}\,\|\mathcal{P}_\Omega(\mathsf{f}_U(\boldsymbol{\alpha}))\|_F^2 \leq T \right\}
$$

and the event

$$
\mathcal{B}_l = \left\{ \exists \boldsymbol{\alpha} \in \tilde{\mathcal{A}}(d_1, \eta^l \nu) : \left| \|\mathcal{P}_\Omega(\mathsf{f}_U(\boldsymbol{\alpha}))\|_F^2 - \mathbb{E}\,\|\mathcal{P}_\Omega(\mathsf{f}_U(\boldsymbol{\alpha}))\|_F^2 \right| > \frac{5}{12}\eta^l \nu + \mathsf{D}_{\boldsymbol{\alpha}} \right\}.
$$

If $\mathcal{B}$ holds, then (43) implies that $\mathcal{B}_l$ holds for some $l \leq 1$. Therefore, $\mathcal{B} \subset \cup_{l=1}^{+\infty}\mathcal{B}_l$, and it is enough to estimate the probability of the events $\mathcal{B}_l$ and then apply the union bound. Such an estimation is given in the following Lemma, adapted from Lemma 10 in [20].

**Lemma 6** *Define* $Z_T = \sup_{\boldsymbol{\alpha} \in \tilde{\mathcal{A}}(d_1, T)} \left| \|\mathcal{P}_\Omega(\mathsf{f}_U(\boldsymbol{\alpha}))\|_F^2 - \mathbb{E}\,\|\mathcal{P}_\Omega(\mathsf{f}_U(\boldsymbol{\alpha}))\|_F^2 \right|$. *Then,*

$$
\mathbb{P}\left( Z_T \geq \mathsf{D}_{\boldsymbol{\alpha}} + \frac{5}{12}T \right) \leq 4\mathrm{e}^{-\pi T/18}.
$$

Lemma 6 gives that $\mathbb{P}(\mathcal{B}_l) \leq 4\exp(-\pi\eta^l\nu/18)$. Applying the union bound we obtain

$$
\begin{aligned}
\mathbb{P}(\mathcal{B}) \quad &\leq \sum_{l=1}^{\infty} \mathbb{P}(\mathcal{B}_l) \\
&\leq 4 \sum_{l=1}^{\infty} \exp(-\pi\eta^l\nu/18) \\
&\leq 4 \sum_{l=1}^{\infty} \exp(-\pi\log(\eta)l\nu/18),
\end{aligned}
$$

where we used $e^x \geq x$. Finally, for $\nu = \frac{18\log(n+p)}{\pi\log(6/5)}$ we obtain

$$
\mathbb{P}(\mathcal{B}) \leq \frac{4\exp(-\pi\nu\log(\eta)/18)}{1 - \exp(-\pi\nu\log(\eta)/18)} \leq \frac{4\exp(-\log(n+p))}{1 - \exp(-\log(n+p))} \leq \frac{8}{n+p},
$$

since $d - 1 \geq (n+p)/2$, which concludes the proof of (i).

**Proof of (ii):** The proof is very similar to that of (i); we recycle some of the notations for simplicity. Recall

$$
\mathsf{D}_X = \frac{112\rho}{\pi}\mathbb{E}\left[\|\Sigma_R\|\right]^2 + 8\nu\varepsilon\mathbb{E}\left[\|\Sigma_R\|\right] + 8\nu d_1 d_{\mathbf{X}}\mathbb{E}\left[\|\Sigma_R\|_\infty\right] + d_\Pi + 768p^{-1}.
$$

Let

$$
\mathcal{B} = \Big\{ \exists (\boldsymbol{\Theta}, \boldsymbol{\alpha}) \in \mathcal{C}(d_1, d_\Pi, \rho, \varepsilon);
$$
$$
\left| \|\mathcal{P}_\Omega(\boldsymbol{\Theta} + \mathsf{f}_U(\boldsymbol{\alpha}))\|_F^2 - \mathbb{E}\,\|\mathcal{P}_\Omega(\boldsymbol{\Theta} + \mathsf{f}_U(\boldsymbol{\alpha}))\|_F^2 \right| > \frac{1}{2}\mathbb{E}\,\|\mathcal{P}_\Omega(\boldsymbol{\Theta} + \mathsf{f}_U(\boldsymbol{\alpha}))\|_F^2 + \mathsf{D}_X \Big\},
$$

$\nu = \frac{72\log(n+p)}{\pi\log(6/5)}$, $\eta = \frac{6}{5}$ and for $l \in \mathbb{N}$

$$
S_l = \left\{ (\boldsymbol{\Theta}, \boldsymbol{\alpha}) \in \mathcal{C}(d_1, d_\Pi, \rho, \varepsilon) : \quad \eta^{l-1}\nu \leq \mathbb{E}\,\|\mathcal{P}_\Omega(\boldsymbol{\Theta} + \mathsf{f}_U(\boldsymbol{\alpha}))\|_F^2 \leq \eta^l \nu \right\}.
$$

As before, if $\mathcal{B}$ holds, then there exist $l \geq 2$ and $(\boldsymbol{\Theta}, \boldsymbol{\alpha}) \in \mathcal{C}(d_1, d_\Pi, \rho, \varepsilon) \cap S_l$ such that

$$\left| \|\mathcal{P}_\Omega(\boldsymbol{\Theta} + \mathsf{f}_U(\boldsymbol{\alpha}))\|_F^2 - \mathbb{E}\|\mathcal{P}_\Omega(\boldsymbol{\Theta} + \mathsf{f}_U(\boldsymbol{\alpha}))\|_F^2 \right| > \frac{5}{12}\eta^l\nu + \mathsf{D}_X. \tag{44}$$

For $T > \nu$, consider the set $\tilde{\mathcal{C}}(T) = \left\{ (\boldsymbol{\Theta}, \boldsymbol{\alpha}) \in \mathcal{C}(d_1, d_\Pi, \rho, \varepsilon) : \mathbb{E}\|\mathcal{P}_\Omega(\boldsymbol{\Theta} + \mathsf{f}_U(\boldsymbol{\alpha}))\|_F^2 \leq T \right\}$, and the event

$$\mathcal{B}_l = \left\{ \exists(\boldsymbol{\Theta}, \boldsymbol{\alpha}) \in \tilde{\mathcal{C}}(\eta^l\nu) : \left| \|\mathcal{P}_\Omega(\boldsymbol{\Theta} + \mathsf{f}_U(\boldsymbol{\alpha}))\|_F^2 - \mathbb{E}\|\mathcal{P}_\Omega(\boldsymbol{\Theta} + \mathsf{f}_U(\boldsymbol{\alpha}))\|_F^2 \right| > \frac{5}{12}\eta^l\nu + \mathsf{D}_X \right\}.$$

Then, (44) implies that $\mathcal{B}_l$ holds and $\mathcal{B} \subset \cup_{l=1}^{+\infty} \mathcal{B}_l$. Thus, we estimate in Lemma 7 the probability of the events $\mathcal{B}_l$, and then apply the union bound.

**Lemma 7** *Let* $W_T = \sup_{(\boldsymbol{\Theta}, \boldsymbol{\alpha}) \in \tilde{\mathcal{C}}(T)} \left| \|\mathcal{P}_\Omega(\boldsymbol{\Theta} + \mathsf{f}_U(\boldsymbol{\alpha}))\|_F^2 - \mathbb{E}\|\mathcal{P}_\Omega(\boldsymbol{\Theta} + \mathsf{f}_U(\boldsymbol{\alpha}))\|_F^2 \right|.$

$$\mathbb{P}\left( W_T \geq \mathsf{D}_X + \frac{5}{12}T \right) \leq 4\mathrm{e}^{-\pi T/72}.$$

Lemma 7 gives that $\mathbb{P}(\mathcal{B}_l) \leq 4\exp(-\pi\eta^l\nu/72)$. Applying the union bound we obtain

$$\begin{aligned}
\mathbb{P}(\mathcal{B}) &\leq \sum_{l=1}^\infty \mathbb{P}(\mathcal{B}_l) \\
&\leq 4\sum_{l=1}^\infty \exp(-\pi\eta^l\nu/72) \\
&\leq 4\sum_{l=1}^\infty \exp(-\pi\log(\eta)l\nu/72),
\end{aligned}$$

where we used $\mathrm{e}^x \geq x$. Finally, for $\nu = \frac{72\log(n+p)}{\pi\log(6/5)}$ we obtain

$$\mathbb{P}(\mathcal{B}) \leq \frac{4\exp(-\pi\nu\log(\eta)/72)}{1 - \exp(-\pi\nu\log(\eta)/72)} \leq \frac{4\exp(-\log(n+p))}{1 - \exp(-\log(n+p))} \leq 8(n+p)^{-1},$$

since $n + p - 1 \geq (n+p)/2$, which concludes the proof of (ii).

## C  Proof of Theorem 2

To prove the theorem, we first lower bound on the progress made by the algorithm at the two blocks between the iterations. With a slight abuse of notations, in the following we shall denote the iterates without the bracket in the superscripts, e.g., we denote $\boldsymbol{\alpha}^{(t)}, \boldsymbol{\Theta}^{(t)}, R^{(t)}$ by $\boldsymbol{\alpha}^t, \boldsymbol{\Theta}^t, R^t$, respectively, to simplify our discussions.

For the first block on $\boldsymbol{\alpha}$, in Section C.1 we show that

$$F(\boldsymbol{\alpha}^t, \boldsymbol{\Theta}^{t-1}, R^{t-1}) \leq F(\boldsymbol{\alpha}^{t-1}, \boldsymbol{\Theta}^{t-1}, R^{t-1}) - \frac{\gamma}{2}\frac{\left(g_{\boldsymbol{\alpha}}(\boldsymbol{\alpha}^{t-1}, \boldsymbol{\Theta}^{t-1}; Q^{t-1})\right)^2}{(2Q^{t-1})^2}, \tag{45}$$

where $Q^{t-1} := \lambda_S^{-1}F(\boldsymbol{\alpha}^{t-1}, \boldsymbol{\Theta}^{t-1}, R^{t-1})$ as defined in the main paper and

$$g_{\boldsymbol{\alpha}}(\boldsymbol{\alpha}^{t-1}, \boldsymbol{\Theta}^{t-1}; Q^{t-1}) := \langle \nabla_{\boldsymbol{\alpha}}\mathcal{L}(\boldsymbol{\alpha}^{t-1}, \boldsymbol{\Theta}^{t-1}), \boldsymbol{\alpha}^{t-1} - \hat{\boldsymbol{\alpha}}^{t-1} \rangle + \lambda_S(\|\boldsymbol{\alpha}^{t-1}\|_1 - \|\hat{\boldsymbol{\alpha}}^{t-1}\|_1), \tag{46}$$

such that

$$\hat{\boldsymbol{\alpha}}^{t-1} := \underset{\boldsymbol{\alpha}}{\arg\min}\ \left(\langle \nabla_{\boldsymbol{\alpha}}\mathcal{L}(\boldsymbol{\alpha}^{t-1}, \boldsymbol{\Theta}^{t-1}), \boldsymbol{\alpha}\rangle + \lambda_S\|\boldsymbol{\alpha}\|_1\right)\ \text{s.t.}\ \|\boldsymbol{\alpha}\|_1 \leq Q^{t-1}. \tag{47}$$

For the second block on $(\boldsymbol{\Theta}, R)$, Section C.2 shows that

$$F(\boldsymbol{\alpha}^t, \boldsymbol{\Theta}^t, R^t) \leq F(\boldsymbol{\alpha}^t, \boldsymbol{\Theta}^{t-1}, R^{t-1}) - \frac{(g_{\boldsymbol{\Theta}}(\boldsymbol{\alpha}^t, \boldsymbol{\Theta}^{t-1}, R^{t-1}; R_{\mathsf{UB}}^t))^2}{\max\{2R_{\mathsf{UB}}^t(\lambda_L + M^t), 8\sigma_{\boldsymbol{\Theta}}(R_{\mathsf{UB}}^t)^2\}}, \tag{48}$$

where $M^t := \|\nabla_{\boldsymbol{\Theta}}(\boldsymbol{\alpha}^t, \boldsymbol{\Theta}^{t-1})\|_2$ and we recall that $R_{\mathsf{UB}}^t := \lambda_L^{-1} F(\boldsymbol{\alpha}^t, \boldsymbol{\Theta}^{t-1}, R^{t-1})$ and we have defined

$$g_{\boldsymbol{\Theta}}(\boldsymbol{\alpha}^t, \boldsymbol{\Theta}^{t-1}, R^{t-1}; R_{\mathsf{UB}}^t) := \langle \boldsymbol{\Theta}^{t-1} - \hat{\boldsymbol{\Theta}}^t, \nabla_{\boldsymbol{\Theta}}\mathcal{L}(\boldsymbol{\alpha}^t, \boldsymbol{\Theta}^{t-1}) \rangle + \lambda_L (R^{t-1} - \hat{R}^t) . \qquad (49)$$

Moreover, Section C.2 shows that

$$F(\boldsymbol{\alpha}^t, \boldsymbol{\Theta}^t, R^t) - F(\boldsymbol{\alpha}^t, \boldsymbol{\Theta}^{t-1}, R^{t-1}) \leq -\frac{\sigma_{\boldsymbol{\Theta}}}{2} \|\mathcal{P}_\Omega(\boldsymbol{\Theta}^t - \boldsymbol{\Theta}^{t-1})\|_F^2 . \qquad (50)$$

**Statement (i).** The above results show that the objective values for the iterates produced by the MCGD method are non-increasing, *i.e.,*

$$F(\boldsymbol{\alpha}^t, \boldsymbol{\Theta}^t, R^t) \leq F(\boldsymbol{\alpha}^t, \boldsymbol{\Theta}^{t-1}, R^{t-1}) \leq F(\boldsymbol{\alpha}^{t-1}, \boldsymbol{\Theta}^{t-1}, R^{t-1}) \qquad (51)$$

Now, consider the time varying part in the quantity $C(t)$ [cf. (18)] — $Q^t, R_{\mathsf{UB}}^t, M^t$. The first two quantities are defined from the objective values and are thus bounded by $\lambda_S^{-1} F(\boldsymbol{\alpha}^0, \boldsymbol{\Theta}^0, R^0)$, $\lambda_L^{-1} F(\boldsymbol{\alpha}^0, \boldsymbol{\Theta}^0, R^0)$, respectively. Moreover, from the monotonicity of $F(\boldsymbol{\alpha}^t, \boldsymbol{\Theta}^{t-1}, R^{t-1})$, we have $\lambda_L \|\boldsymbol{\Theta}^{t-1}\|_\star + \lambda_S \|\boldsymbol{\alpha}^t\|_1 \leq F(\boldsymbol{\alpha}^t, \boldsymbol{\Theta}^{t-1}, R^{t-1}) \leq F(\boldsymbol{\alpha}^0, \boldsymbol{\Theta}^0, R^0)$ for all $t \geq 1$. As the gradient $\nabla_{\boldsymbol{\Theta}}\mathcal{L}(\boldsymbol{\alpha}, \boldsymbol{\Theta})$ is bounded whenever $\boldsymbol{\alpha}, \boldsymbol{\Theta}$ are bounded, we conclude that $M^t$ is bounded, e.g., $M^t \leq \bar{M}$. Finally, this shows for all $t \geq 1$ that

$$C(t) \leq \overline{C} := \max \left\{ \frac{24(Q^0)^2}{\gamma}, \frac{24\hat{\sigma}_{\boldsymbol{\Theta}}^2 (Q^0)^2}{\sigma_{\boldsymbol{\Theta}}} + \max\{6 R_{\mathsf{UB}}^0 (\lambda_L + \bar{M}), 24\sigma_{\boldsymbol{\Theta}} (R_{\mathsf{UB}}^0)^2\} \right\} . \qquad (52)$$

**Statement (ii).** To characterize the convergence rate of the MCGD method, let us consider the Lyapunov function, $g^t(Q^t, R_{\mathsf{UB}}^t)$, defined as:

$$g^t(Q^t, R_{\mathsf{UB}}^t) := g_{\boldsymbol{\alpha}}(\boldsymbol{\alpha}^t, \boldsymbol{\Theta}^{t-1}; Q^t) + g_{\boldsymbol{\Theta}}(\boldsymbol{\alpha}^t, \boldsymbol{\Theta}^{t-1}, R^{t-1}; R_{\mathsf{UB}}^t) . \qquad (53)$$

Note that as the loss function $\mathcal{L}(\boldsymbol{\alpha}, \boldsymbol{\Theta})$ is convex and $\|\hat{\boldsymbol{\alpha}}\|_1 \leq Q^t, \|\hat{\boldsymbol{\Theta}}\|_\star \leq R_{\mathsf{UB}}^t$, it is possible to lower bound $g^t(Q^t, R_{\mathsf{UB}}^t)$ by:

$$g^t(Q^t, R_{\mathsf{UB}}^t) \geq F(\boldsymbol{\alpha}^t, \boldsymbol{\Theta}^{t-1}, R^{t-1}) - F(\hat{\boldsymbol{\alpha}}, \hat{\boldsymbol{\Theta}}, \hat{R}) . \qquad (54)$$

Furthermore, we can obtain an upper bound to $g^t(Q^t, R_{\mathsf{UB}}^t)$ in terms of the objective values:

$$
\begin{aligned}
g_{\boldsymbol{\alpha}}(\boldsymbol{\alpha}^t, \boldsymbol{\Theta}^{t-1}; Q^t) &= \max_{\|\boldsymbol{\alpha}\|_1 \leq Q^t} \langle \nabla_{\boldsymbol{\alpha}}\mathcal{L}(\boldsymbol{\alpha}^t, \boldsymbol{\Theta}^{t-1}), \boldsymbol{\alpha}^t - \boldsymbol{\alpha} \rangle + \lambda_S(\|\boldsymbol{\alpha}^t\|_1 - \|\boldsymbol{\alpha}\|_1) \\
&= \max_{\|\boldsymbol{\alpha}\|_1 \leq Q^t} \langle \nabla_{\boldsymbol{\alpha}}\mathcal{L}(\boldsymbol{\alpha}^t, \boldsymbol{\Theta}^t), \boldsymbol{\alpha}^t - \boldsymbol{\alpha} \rangle + \langle \nabla_{\boldsymbol{\alpha}}\mathcal{L}(\boldsymbol{\alpha}^t, \boldsymbol{\Theta}^{t-1}) - \nabla_{\boldsymbol{\alpha}}\mathcal{L}(\boldsymbol{\alpha}^t, \boldsymbol{\Theta}^t), \boldsymbol{\alpha}^t - \boldsymbol{\alpha} \rangle \\
&\qquad + \lambda_S(\|\boldsymbol{\alpha}^t\|_1 - \|\boldsymbol{\alpha}\|_1) \\
&\leq \max_{\|\boldsymbol{\alpha}\|_1 \leq Q^t} \langle \nabla_{\boldsymbol{\alpha}}\mathcal{L}(\boldsymbol{\alpha}^t, \boldsymbol{\Theta}^t), \boldsymbol{\alpha}^t - \boldsymbol{\alpha} \rangle + \lambda_S(\|\boldsymbol{\alpha}^t\|_1 - \|\boldsymbol{\alpha}\|_1) \\
&\qquad + \|\nabla_{\boldsymbol{\alpha}}\mathcal{L}(\boldsymbol{\alpha}^t, \boldsymbol{\Theta}^{t-1}) - \nabla_{\boldsymbol{\alpha}}\mathcal{L}(\boldsymbol{\alpha}^t, \boldsymbol{\Theta}^t)\|_2 \|\boldsymbol{\alpha}^t - \boldsymbol{\alpha}\|_2 \\
&\leq g_{\boldsymbol{\alpha}}(\boldsymbol{\alpha}^t, \boldsymbol{\Theta}^t; Q^t) + 2\hat{\sigma}_{\boldsymbol{\Theta}} Q^t \|\mathcal{P}_\Omega(\boldsymbol{\Theta}^{t-1} - \boldsymbol{\Theta}^t)\|_F .
\end{aligned}
\qquad (55)
$$

Consequently, we have

$$
\begin{aligned}
&\big(g^t(Q^t, R_{\mathsf{UB}}^t)\big)^2 \\
&\leq 3\big((g_{\boldsymbol{\Theta}}(\boldsymbol{\alpha}^t, \boldsymbol{\Theta}^{t-1}, R^{t-1}; R_{\mathsf{UB}}^t))^2 + (g_{\boldsymbol{\alpha}}(\boldsymbol{\alpha}^t, \boldsymbol{\Theta}^t; Q^t))^2 + 4\hat{\sigma}_{\boldsymbol{\Theta}}^2 (Q^t)^2 \|\mathcal{P}_\Omega(\boldsymbol{\Theta}^{t-1} - \boldsymbol{\Theta}^t)\|_F^2 \big) \\
&\leq 3\big(C_1^t \big(F(\boldsymbol{\alpha}^t, \boldsymbol{\Theta}^t, R^t) - F(\boldsymbol{\alpha}^{t+1}, \boldsymbol{\Theta}^t, R^t)\big) + C_2^t \big(F(\boldsymbol{\alpha}^t, \boldsymbol{\Theta}^{t-1}, R^{t-1}) - F(\boldsymbol{\alpha}^t, \boldsymbol{\Theta}^t, R^t)\big)\big)
\end{aligned}
$$

where

$$C_1^t := \frac{8(Q^t)^2}{\gamma}, \quad C_2^t = \frac{8\hat{\sigma}_{\boldsymbol{\Theta}}^2 (Q^t)^2}{\sigma_{\boldsymbol{\Theta}}} + \max\{2 R_{\mathsf{UB}}^t (\lambda_L + M), 8\sigma_{\boldsymbol{\Theta}} (R_{\mathsf{UB}}^t)^2\} \qquad (56)$$

Observe that $C(t)$ is defined by $C(t) = 3\max\{C_1^t, C_2^t\}$ as the upper bound of the above constants, we get

$$\big(g^t(Q^t, R_{\mathsf{UB}}^t)\big)^2 \leq C(t)\big(F(\boldsymbol{\alpha}^t, \boldsymbol{\Theta}^{t-1}, R^{t-1}) - F(\boldsymbol{\alpha}^{t+1}, \boldsymbol{\Theta}^t, R^t)\big) . \qquad (57)$$

Using the shorthand notation $\Delta^t := F(\boldsymbol{\alpha}^t, \boldsymbol{\Theta}^{t-1}, R^{t-1}) - F(\hat{\boldsymbol{\alpha}}, \hat{\boldsymbol{\Theta}}, \hat{R})$ and notice that $\left(g^t(Q^t, R^t_{\mathsf{UB}})\right)^2 \geq (\Delta^t)^2$, we arrive at the following inequality:

$$\Delta^{t+1} \leq \Delta^t - \frac{1}{C(t)}(\Delta^t)^2 \tag{58}$$

Applying Lemma 8 in Section C.3, we can show that

$$\Delta^{t+1} \leq \frac{1}{(\Delta^1)^{-1} + \sum_{i=1}^t \frac{1}{C(i)}} \ , \tag{59}$$

Note that $\Delta^1 \leq \tilde{\Delta}^0 := F(\boldsymbol{\alpha}^0, \boldsymbol{\Theta}^0, R^0) - F(\hat{\boldsymbol{\alpha}}, \hat{\boldsymbol{\Theta}}, \hat{R})$, we have

$$\Delta^{t+1} \leq \frac{1}{(\tilde{\Delta}^0)^{-1} + \sum_{i=1}^t \frac{1}{C(i)}} \leq \frac{1}{(\tilde{\Delta}^0)^{-1} + t\overline{C}(t)}, \ \forall\, t \geq 0 \ . \tag{60}$$

The proof is concluded by the straightforward inequality $F_0(\boldsymbol{\alpha}^{t+1}, \boldsymbol{\Theta}^{t+1}) - F_0(\hat{\boldsymbol{\alpha}}, \hat{\boldsymbol{\Theta}}) \leq F(\boldsymbol{\alpha}^{t+1}, \boldsymbol{\Theta}^{t+1}, R^{t+1}) - F(\hat{\boldsymbol{\alpha}}, \hat{\boldsymbol{\Theta}}, \hat{R}) \leq \Delta^{t+1}$.

**Comment on** $\lim_{t\to\infty} C(t)$. Since both $F(\boldsymbol{\alpha}^t, \boldsymbol{\Theta}^t, R^t)$ and $F(\boldsymbol{\alpha}^t, \boldsymbol{\Theta}^{t-1}, R^{t-1})$ converge to $F^\star := F(\hat{\boldsymbol{\alpha}}, \hat{\boldsymbol{\Theta}}, \hat{R})$, *i.e.,* the optimal objective value. It is clear that $Q^t \to \hat{Q} := \lambda_S^{-1} F^\star$ and $R^t_{\mathsf{UB}} \to \hat{R}_{\mathsf{UB}} := \lambda_L^{-1} F^\star$ as well. Furthermore, by continuity of the gradient, we have $M^t \to \|\nabla_{\boldsymbol{\Theta}}\mathcal{L}(\hat{\boldsymbol{\alpha}}, \hat{\boldsymbol{\Theta}})\|_2$. This shows that the limit $C^\star = \lim_{t\to\infty} C(t)$ exists.

To obtain a computable bound for $C^\star$, note that $(\hat{\boldsymbol{\alpha}}, \hat{\boldsymbol{\Theta}})$ is also an optimal solution to (5) and the optimality condition shows that

$$\mathbf{0} \in \nabla_{\boldsymbol{\Theta}}\mathcal{L}(\hat{\boldsymbol{\alpha}}, \hat{\boldsymbol{\Theta}}) + \lambda_L \partial\|\hat{\boldsymbol{\Theta}}\|_\star \tag{61}$$

By [32, P. 41], we know that $\partial\|\hat{\boldsymbol{\Theta}}\|_\star = \{\boldsymbol{U}_1\boldsymbol{V}_1^\top + \boldsymbol{W} \ : \ \|\boldsymbol{W}\|_2 \leq 1, \ \boldsymbol{U}_1^\top\boldsymbol{W} = \mathbf{0}, \ \boldsymbol{W}\boldsymbol{V}_1 = \mathbf{0}\}$ such that $\boldsymbol{U}_1 \in \mathbb{R}^{m_1 \times r}, \boldsymbol{V}_1 \in \mathbb{R}^{m_2 \times r}$ are the left/right singular vectors of $\hat{\boldsymbol{\Theta}}$ corresponding the $r := \mathrm{rank}(\hat{\boldsymbol{\Theta}})$ non-zero singular values of $\hat{\boldsymbol{\Theta}}$. Importantly, this implies that $\|\nabla_{\boldsymbol{\Theta}}\mathcal{L}(\hat{\boldsymbol{\alpha}}, \hat{\boldsymbol{\Theta}})\|_2 \leq 2\lambda_L$ and

$$C^\star \leq \overline{C}^\star := \max\left\{\frac{24(\hat{Q})^2}{\gamma}, \frac{24\hat{\sigma}_{\boldsymbol{\Theta}}^2(\hat{Q})^2}{\sigma_{\boldsymbol{\Theta}}} + \max\{18\hat{R}_{\mathsf{UB}}\lambda_L, 24\sigma_{\boldsymbol{\Theta}}(\hat{R}_{\mathsf{UB}})^2\}\right\} \ . \tag{62}$$

### C.1 Proof of Eq. (45)

Suppose $\boldsymbol{\alpha}^t$ is obtained by the proximal update in (10), we observe that

$$F(\boldsymbol{\alpha}^t, \boldsymbol{\Theta}^{t-1}, R^{t-1}) \leq F(\boldsymbol{\alpha}^{t-1}, \boldsymbol{\Theta}^{t-1}, R^{t-1}) + \langle\nabla_{\boldsymbol{\alpha}}\mathcal{L}(\boldsymbol{\alpha}^{t-1}, \boldsymbol{\Theta}^{t-1}), \boldsymbol{\alpha}^t - \boldsymbol{\alpha}^{t-1}\rangle$$
$$+ \frac{\sigma_{\boldsymbol{\alpha}}}{2}\|\boldsymbol{\alpha}^t - \boldsymbol{\alpha}^{t-1}\|_2^2 + \lambda_S\left(\|\boldsymbol{\alpha}^t\|_1 - \|\boldsymbol{\alpha}^{t-1}\|_1\right) \ . \tag{63}$$

On the other hand, when $\boldsymbol{\alpha}^t$ is obtained by the exact minimization in (11), denoted by $\boldsymbol{\alpha}^t_{\mathsf{exact}}$ to avoid confusion, we have $F(\boldsymbol{\alpha}^t_{\mathsf{exact}}, \boldsymbol{\Theta}^{t-1}, R^{t-1}) \leq F(\boldsymbol{\alpha}^t, \boldsymbol{\Theta}^{t-1}, R^{t-1})$ since the latter is an exact minimizer. Thus, $F(\boldsymbol{\alpha}^t_{\mathsf{exact}}, \boldsymbol{\Theta}^{t-1}, R^{t-1})$ is upper bounded by the right hand side in the above inequality.

Using the property of the proximal operator, it can be shown that

$$\boldsymbol{\alpha}^t \in \underset{\boldsymbol{\alpha}}{\arg\min} \left(\langle\nabla_{\boldsymbol{\alpha}}\mathcal{L}(\boldsymbol{\alpha}^{t-1}, \boldsymbol{\Theta}^{t-1}), \boldsymbol{\alpha} - \boldsymbol{\alpha}^{t-1}\rangle + \frac{1}{2\gamma}\|\boldsymbol{\alpha} - \boldsymbol{\alpha}^{t-1}\|_2^2 + \lambda_S(\|\boldsymbol{\alpha}\|_1 - \|\boldsymbol{\alpha}^{t-1}\|_1)\right) \tag{64}$$

Due to our choice of step size, we have $\sigma_{\boldsymbol{\alpha}} \leq 1/\gamma$. Combining this with the above inequality implies that

$$F(\boldsymbol{\alpha}^t, \boldsymbol{\Theta}^{t-1}, R^{t-1}) \leq F(\boldsymbol{\alpha}^{t-1}, \boldsymbol{\Theta}^{t-1}, R^{t-1}) + \langle\nabla_{\boldsymbol{\alpha}}\mathcal{L}(\boldsymbol{\alpha}^{t-1}, \boldsymbol{\Theta}^{t-1}), \boldsymbol{\alpha} - \boldsymbol{\alpha}^{t-1}\rangle$$
$$+ \frac{1}{2\gamma}\|\boldsymbol{\alpha} - \boldsymbol{\alpha}^{t-1}\|_2^2 + \lambda_S\left(\|\boldsymbol{\alpha}\|_1 - \|\boldsymbol{\alpha}^{t-1}\|_1\right), \ \forall\, \boldsymbol{\alpha} \in \mathbb{R}^K \ . \tag{65}$$

Furthermore, for all $b \in \mathbb{R}$ it holds that

$$
\begin{aligned}
F(\boldsymbol{\alpha}^t, \boldsymbol{\Theta}^{t-1}, R^{t-1}) &\leq F(\boldsymbol{\alpha}^{t-1}, \boldsymbol{\Theta}^{t-1}, R^{t-1}) + b\langle \nabla_{\boldsymbol{\alpha}} \mathcal{L}(\boldsymbol{\alpha}^{t-1}, \boldsymbol{\Theta}^{t-1}), \hat{\boldsymbol{\alpha}}^{t-1} - \boldsymbol{\alpha}^{t-1}\rangle \\
&\quad + \frac{b^2}{2\gamma}\|\hat{\boldsymbol{\alpha}}^{t-1} - \boldsymbol{\alpha}^{t-1}\|_2^2 + \lambda_S\big(\|b\hat{\boldsymbol{\alpha}}^{t-1} + (1-b)\boldsymbol{\alpha}^{t-1}\|_1 - \|\boldsymbol{\alpha}^{t-1}\|_1\big) \\
&\leq F(\boldsymbol{\alpha}^{t-1}, \boldsymbol{\Theta}^{t-1}, R^{t-1}) + b\langle \nabla_{\boldsymbol{\alpha}} \mathcal{L}(\boldsymbol{\alpha}^{t-1}, \boldsymbol{\Theta}^{t-1}), \hat{\boldsymbol{\alpha}}^{t-1} - \boldsymbol{\alpha}^{t-1}\rangle \\
&\quad + \frac{b^2}{2\gamma}\|\hat{\boldsymbol{\alpha}}^{t-1} - \boldsymbol{\alpha}^{t-1}\|_2^2 + b\lambda_S\big(\|\hat{\boldsymbol{\alpha}}^{t-1}\|_1 - \|\boldsymbol{\alpha}^{t-1}\|_1\big) \,,
\end{aligned} \tag{66}
$$

where we have limited our search space from $\boldsymbol{\alpha} \in \mathbb{R}^K$ to $\boldsymbol{\alpha} = b\hat{\boldsymbol{\alpha}}^t + (1-b)\boldsymbol{\alpha}^{t-1}$ for $b \in \mathbb{R}$. Minimizing the right hand side of the above with respect to $b$ yields

$$
\begin{aligned}
&F(\boldsymbol{\alpha}^t, \boldsymbol{\Theta}^{t-1}, R^{t-1}) - F(\boldsymbol{\alpha}^{t-1}, \boldsymbol{\Theta}^{t-1}, R^{t-1}) \\
&\leq -\frac{\gamma}{2}\frac{(g_{\boldsymbol{\alpha}}(\boldsymbol{\alpha}^{t-1}, \boldsymbol{\Theta}^{t-1}; Q^{t-1}))^2}{\|\hat{\boldsymbol{\alpha}}^{t-1} - \boldsymbol{\alpha}^{t-1}\|_2^2} \leq -\frac{\gamma}{2}\frac{(g_{\boldsymbol{\alpha}}(\boldsymbol{\alpha}^{t-1}, \boldsymbol{\Theta}^{t-1}; Q^{t-1}))^2}{(2Q^{t-1})^2} \,,
\end{aligned} \tag{67}
$$

where we have used $\|\hat{\boldsymbol{\alpha}}^{t-1} - \boldsymbol{\alpha}^{t-1}\|_2^2 \leq (2Q^{t-1})^2$ in the last inequality.

## C.2  Proof of Eq. (48) and (50)

Let us observe that

$$
\begin{aligned}
F(\boldsymbol{\alpha}^t, \boldsymbol{\Theta}^t, R^t) &= F(\boldsymbol{\alpha}^t, \boldsymbol{\Theta}^{t-1}, R^{t-1}) - \beta_t g_{\boldsymbol{\Theta}}(\boldsymbol{\alpha}^t, \boldsymbol{\Theta}^{t-1}, R^{t-1}; R_{\mathsf{UB}}^t) \\
&\quad + \frac{\beta_t^2}{2}\begin{pmatrix} \mathrm{vec}(\hat{\boldsymbol{\Theta}}^t - \boldsymbol{\Theta}^{t-1}) \\ \hat{R}^t - R^{t-1} \end{pmatrix}^\top \nabla_{\boldsymbol{\Theta},R}^2(\boldsymbol{\xi}) \begin{pmatrix} \mathrm{vec}(\hat{\boldsymbol{\Theta}}^t - \boldsymbol{\Theta}^{t-1}) \\ \hat{R}^t - R^{t-1} \end{pmatrix} \,,
\end{aligned} \tag{68}
$$

where $\boldsymbol{\xi}$ is any point that lies on the line $[(\mathrm{vec}(\boldsymbol{\Theta}^{t-1}); R^{t-1}), (\mathrm{vec}(\boldsymbol{\Theta}^t); R^t)]$. From the property of $F$, we observe that

$$
\nabla_{\boldsymbol{\Theta},R}^2(\boldsymbol{\xi}) \preceq \begin{pmatrix} \sigma_{\boldsymbol{\Theta}}\mathrm{Diag}(\mathcal{P}_{\Omega}(\mathbf{J})) & \mathbf{0} \\ \mathbf{0} & \mathbf{0} \end{pmatrix} \,, \tag{69}
$$

where $\mathbf{J}$ is the $m_1 \times m_2$ all-ones matrix. The above implies that

$$
\begin{aligned}
F(\boldsymbol{\alpha}^t, \boldsymbol{\Theta}^t, R^t) &\leq F(\boldsymbol{\alpha}^t, \boldsymbol{\Theta}^{t-1}, R^{t-1}) - \beta_t g_{\boldsymbol{\Theta}}(\boldsymbol{\alpha}^t, \boldsymbol{\Theta}^{t-1}, R^{t-1}; R_{\mathsf{UB}}^t) \\
&\quad + \frac{\beta_t^2 \sigma_{\boldsymbol{\Theta}}}{2}\|\mathcal{P}_{\Omega}(\hat{\boldsymbol{\Theta}}^t - \boldsymbol{\Theta}^{t-1})\|_F^2 \,.
\end{aligned} \tag{70}
$$

Recall that $\beta_t = \min\{1, g_{\boldsymbol{\Theta}}(\boldsymbol{\alpha}^t, \boldsymbol{\Theta}^{t-1}, R^{t-1}; R_{\mathsf{UB}}^t)/(\sigma_{\boldsymbol{\Theta}}\|\mathcal{P}_{\Omega}(\hat{\boldsymbol{\Theta}}^t - \boldsymbol{\Theta}^{t-1})\|_F^2)\}$. If $g_{\boldsymbol{\Theta}}(\boldsymbol{\alpha}^t, \boldsymbol{\Theta}^{t-1}, R^{t-1}; R_{\mathsf{UB}}^t) \geq \sigma_{\boldsymbol{\Theta}}\|\mathcal{P}_{\Omega}(\hat{\boldsymbol{\Theta}}^t - \boldsymbol{\Theta}^{t-1})\|_F^2$, then we choose $\beta_t = 1$ and observe:

$$
\begin{aligned}
&F(\boldsymbol{\alpha}^t, \boldsymbol{\Theta}^t, R^t) - F(\boldsymbol{\alpha}^t, \boldsymbol{\Theta}^{t-1}, R^{t-1}) \\
&\leq -\frac{1}{2} g_{\boldsymbol{\Theta}}(\boldsymbol{\alpha}^t, \boldsymbol{\Theta}^{t-1}, R^{t-1}; R_{\mathsf{UB}}^t) = -\frac{1}{2}\frac{(g_{\boldsymbol{\Theta}}(\boldsymbol{\alpha}^t, \boldsymbol{\Theta}^{t-1}, R^{t-1}; R_{\mathsf{UB}}^t))^2}{g_{\boldsymbol{\Theta}}(\boldsymbol{\alpha}^t, \boldsymbol{\Theta}^{t-1}, R^{t-1}; R_{\mathsf{UB}}^t)} \\
&\leq -\frac{1}{2}\frac{(g_{\boldsymbol{\Theta}}(\boldsymbol{\alpha}^t, \boldsymbol{\Theta}^{t-1}, R^{t-1}; R_{\mathsf{UB}}^t))^2}{R_{\mathsf{UB}}^t(\lambda_L + 2M^t)} \,,
\end{aligned} \tag{71}
$$

where we have used the upper bound to $g_{\boldsymbol{\Theta}}(\boldsymbol{\alpha}^t, \boldsymbol{\Theta}^{t-1}, R^{t-1}; R_{\mathsf{UB}}^t)$ as follows:

$$
\begin{aligned}
g_{\boldsymbol{\Theta}}(\boldsymbol{\alpha}^t, \boldsymbol{\Theta}^{t-1}, R^{t-1}; R_{\mathsf{UB}}^t) &\leq \lambda_L R_{\mathsf{UB}}^t + \langle \boldsymbol{\Theta}^{t-1} - \hat{\boldsymbol{\Theta}}^t, \nabla_{\boldsymbol{\Theta}}\mathcal{L}(\boldsymbol{\alpha}^t, \boldsymbol{\Theta}^{t-1})\rangle \\
&\leq R_{\mathsf{UB}}^t(\lambda_L + 2M^t) \,,
\end{aligned} \tag{72}
$$

with $M^t := \|\nabla_{\boldsymbol{\Theta}}\mathcal{L}(\boldsymbol{\alpha}^t, \boldsymbol{\Theta}^{t-1})\|_2$ being the spectral norm of the gradient.

Otherwise, we choose $\beta_t = g_{\boldsymbol{\Theta}}(\boldsymbol{\alpha}^t, \boldsymbol{\Theta}^{t-1}, R^{t-1}; R_{\mathsf{UB}}^t)/(\sigma_{\boldsymbol{\Theta}}\|\mathcal{P}_{\Omega}(\hat{\boldsymbol{\Theta}}^t - \boldsymbol{\Theta}^{t-1})\|_F^2)$ and observe:

$$
\begin{aligned}
&F(\boldsymbol{\alpha}^t, \boldsymbol{\Theta}^t, R^t) - F(\boldsymbol{\alpha}^t, \boldsymbol{\Theta}^{t-1}, R^{t-1}) \\
&\leq -\frac{1}{2}\frac{(g_{\boldsymbol{\Theta}}(\boldsymbol{\alpha}^t, \boldsymbol{\Theta}^{t-1}, R^{t-1}; R_{\mathsf{UB}}^t))^2}{\sigma_{\boldsymbol{\Theta}}\|\mathcal{P}_{\Omega}(\hat{\boldsymbol{\Theta}}^t - \boldsymbol{\Theta}^{t-1})\|_F^2} \leq -\frac{1}{2}\frac{(g_{\boldsymbol{\Theta}}(\boldsymbol{\alpha}^t, \boldsymbol{\Theta}^{t-1}, R^{t-1}; R_{\mathsf{UB}}^t))^2}{\sigma_{\boldsymbol{\Theta}}(2R_{\mathsf{UB}}^t)^2} \,,
\end{aligned} \tag{73}
$$

where we have used $\|\mathcal{P}_\Omega(\hat{\boldsymbol{\Theta}}^t - \boldsymbol{\Theta}^{t-1})\|_F^2 \leq \|\hat{\boldsymbol{\Theta}}^t - \boldsymbol{\Theta}^{t-1}\|_F^2 \leq \|\hat{\boldsymbol{\Theta}}^t - \boldsymbol{\Theta}^{t-1}\|_\star^2 \leq (2R_{\mathsf{UB}}^t)^2$.

To prove (50), we observe that

$$\|\mathcal{P}_\Omega(\boldsymbol{\Theta}^t - \boldsymbol{\Theta}^{t-1})\|_F^2 = \beta_t^2 \|\mathcal{P}_\Omega(\hat{\boldsymbol{\Theta}}^t - \boldsymbol{\Theta}^{t-1})\|_F^2 \ . \tag{74}$$

If $\beta_t = 1$, then we have $g_{\boldsymbol{\Theta}}(\boldsymbol{\alpha}^t, \boldsymbol{\Theta}^{t-1}, R^{t-1}; R_{\mathsf{UB}}^t) \geq \sigma_{\boldsymbol{\Theta}} \|\mathcal{P}_\Omega(\hat{\boldsymbol{\Theta}}^t - \boldsymbol{\Theta}^{t-1})\|_F^2$ and therefore we can upper bound $\|\mathcal{P}_\Omega(\boldsymbol{\Theta}^t - \boldsymbol{\Theta}^{t-1})\|_F^2$ by:

$$\frac{1}{\sigma_{\boldsymbol{\Theta}}} g_{\boldsymbol{\Theta}}(\boldsymbol{\alpha}^t, \boldsymbol{\Theta}^{t-1}, R^{t-1}; R_{\mathsf{UB}}^t) \leq \frac{2}{\sigma_{\boldsymbol{\Theta}}} \Big( F(\boldsymbol{\alpha}^t, \boldsymbol{\Theta}^{t-1}, R^{t-1}) - F(\boldsymbol{\alpha}^t, \boldsymbol{\Theta}^t, R^t) \Big) \tag{75}$$

where the last inequality follows from (72). Otherwise, we choose $\beta_t = g_{\boldsymbol{\Theta}}(\boldsymbol{\alpha}^t, \boldsymbol{\Theta}^{t-1}, R^{t-1}; R_{\mathsf{UB}}^t)/\sigma_{\boldsymbol{\Theta}} \|\mathcal{P}_\Omega(\hat{\boldsymbol{\Theta}}^t - \boldsymbol{\Theta}^{t-1})\|_F^2$ and therefore,

$$\begin{aligned}
\|\mathcal{P}_\Omega(\boldsymbol{\Theta}^t - \boldsymbol{\Theta}^{t-1})\|_F^2 &= \frac{1}{\sigma_{\boldsymbol{\Theta}}} \frac{(g_{\boldsymbol{\Theta}}(\boldsymbol{\alpha}^t, \boldsymbol{\Theta}^{t-1}, R^{t-1}; R_{\mathsf{UB}}^t))^2}{\sigma_{\boldsymbol{\Theta}} \|\mathcal{P}_\Omega(\hat{\boldsymbol{\Theta}}^t - \boldsymbol{\Theta}^{t-1})\|_F^2} \\
&\leq \frac{2}{\sigma_{\boldsymbol{\Theta}}} \Big( F(\boldsymbol{\alpha}^t, \boldsymbol{\Theta}^{t-1}, R^{t-1}) - F(\boldsymbol{\alpha}^t, \boldsymbol{\Theta}^t, R^t) \Big) \ ,
\end{aligned} \tag{76}$$

where the last inequality follows from (73).

### C.3  Additional Lemma

The following lemma is modified from [3, Lemma 3.5].

**Lemma 8** *Let $\{A_k\}_{k \geq 1}$ be a non-negative sequence satisfying:*

$$A_{k+1} \leq A_k - \gamma_k A_k^2, \ k \geq 1 \ , \tag{77}$$

*where $\gamma_k$ is some positive number for all $k \geq 1$. Then,*

$$A_{k+1} \leq \frac{1}{\frac{1}{A_1} + \sum_{i=1}^k \gamma_i}, \ k \geq 1 \ . \tag{78}$$

*Proof*: Consider the following chain of inequality:

$$\frac{1}{A_{k+1}} - \frac{1}{A_k} = \frac{A_k - A_{k+1}}{A_k A_{k+1}} \geq \gamma_k \frac{A_k}{A_{k+1}} \geq \gamma_k \ , \tag{79}$$

where the last inequality is due to the fact that $A_{k+1} \leq A_k$. Consequently, we have

$$\frac{1}{A_{k+1}} - \frac{1}{A_1} = \sum_{i=1}^k \Big( \frac{1}{A_{i+1}} - \frac{1}{A_i} \Big) \geq \sum_{i=1}^k \gamma_i \ . \tag{80}$$

Reshuffling terms shows the desired result in (78).                    **Q.E.D.**

## D  Distributed MCGD Optimization

Similar to the previous section, in the following we shall denote the iterates without the bracket in the superscripts, e.g., we denote $\boldsymbol{\alpha}^{(t)}, \boldsymbol{\Theta}^{(t)}, R^{(t)}$ by $\boldsymbol{\alpha}^t, \boldsymbol{\Theta}^t, R^t$, respectively, to simplify our discussions.

Let us describe a distributed version of the MCGD method under a *master-slave* architecture setting where there exists $K$ workers and each of them is connected to a central server. Our goal is to offload the computation required by MCGD method to the workers, while protecting the privacy sensitive data owned by the workers. To describe our setting, the set of observed data $\mathbf{Y}_{ij}$, $(i,j) \in \Omega$ are stored in $K$ different workers, where the $k$th worker holds $\mathbf{Y}_{ij}$ with $(i,j) \in \Omega_k \subset \Omega$. Particularly, we have $\Omega = \Omega_1 \cup \cdots \cup \Omega_K$ with $\Omega_k \cap \Omega_{k'} = \emptyset$ for all $k \neq k'$. In this way, we can write

$$\mathcal{L}(\boldsymbol{\alpha}, \boldsymbol{\Theta}) = \sum_{(i,j) \in \Omega} \{-\mathbf{Y}_{ij}\mathbf{M}_{ij} + g_j(\mathbf{M}_{ij})\} = \sum_{k=1}^K \underbrace{\sum_{(i,j) \in \Omega_k} \{-\mathbf{Y}_{ij}\mathbf{M}_{ij} + g_j(\mathbf{M}_{ij})\}}_{:=\mathcal{L}_k(\boldsymbol{\alpha}, \boldsymbol{\Theta})} \tag{81}$$

such that the log-likelihood function can be decomposed as $\mathcal{L}(\boldsymbol{\alpha}, \boldsymbol{\Theta}) := \sum_{k=1}^{K} \mathcal{L}_k(\boldsymbol{\alpha}, \boldsymbol{\Theta})$. Moreover, notice that $\mathcal{L}_k(\boldsymbol{\alpha}, \mathcal{P}_{\Omega_k}(\boldsymbol{\Theta})) = \mathcal{L}_k(\boldsymbol{\alpha}, \boldsymbol{\Theta})$ since the $k$th local function is evaluated only on the entries in $\Omega_k$. For simplicity, we assume that computation can be done synchronously among the workers.

We can implement the MCGD method in a distributed setting as follows. We focus on the $t$th iteration where $\boldsymbol{\alpha}^{t-1}, \boldsymbol{\Theta}^{t-1}, R^{t-1}$ have been previously computed and worker $k$ now holds $\boldsymbol{\alpha}^{t-1}, \mathcal{P}_{\Omega_k}(\boldsymbol{\Theta}^{t-1}), R^{t-1}$.

Firstly, the proximal update step of line 3 is replaced by a natural distributed implementation where the workers compute and transmit the local gradients of the log-likelihood function, $\nabla_{\boldsymbol{\alpha}} \mathcal{L}_k(\boldsymbol{\alpha}^{t-1}, \mathcal{P}_{\Omega_k}(\boldsymbol{\Theta}^{t-1}))$, to the master node; the master node can then *aggregate* the received local gradients to form the update in (10), yielding $\boldsymbol{\alpha}^t$ which is then transmitted back to the workers.

Secondly, the CG update of line 5 requires the top SVD of $\nabla_{\boldsymbol{\Theta}} \mathcal{L}(\boldsymbol{\alpha}^t, \boldsymbol{\Theta}^{t-1})$ whose complexity is $\mathcal{O}(|\Omega| \max\{n, p\} \log(1/\delta))$ using a centralized implementation, where $\delta > 0$ is the desired accuracy of SVD. In a distributed setting, we can replace the step by a *distributed power method* for offloading the complexity. Importantly, we observe that the top singular vectors of $\nabla_{\boldsymbol{\Theta}} \mathcal{L}(\boldsymbol{\alpha}^t, \boldsymbol{\Theta}^{t-1})$ can be *approximated* by the following power method recursions:

---

**Algorithm 2** Distributed Power Method for MCGD.

---

1: **Initialize:** initialization — $\boldsymbol{u}(0) \sim \mathcal{N}(\boldsymbol{0}, \boldsymbol{I}) \in \mathbb{R}^n$, and the parameter $P \in \mathbb{Z}$.
2: **for** $p = 1, 2, \ldots, P$ **do**
3:     The central server sends the vector $\boldsymbol{u}(p-1)$ to workers.
4:     For all $k$, worker $k$ computes the vector:

$$\boldsymbol{v}_k(p) = \nabla_{\boldsymbol{\Theta}} \mathcal{L}_k(\boldsymbol{\alpha}^t, \mathcal{P}_{\Omega_k}(\boldsymbol{\Theta}^{t-1})) \boldsymbol{u}(p-1) \tag{82}$$

    and transmit it to the central server.
5:     The central server forms the next iterate by $\boldsymbol{v}(p) = \sum_{k=1}^{K} \boldsymbol{v}_k(p)$ and sends the vector $\boldsymbol{v}(p)$ to workers.
6:     For all $k$, worker $k$ computes the vector:

$$\boldsymbol{u}_k(p) = \nabla_{\boldsymbol{\Theta}} \mathcal{L}_k(\boldsymbol{\alpha}^t, \mathcal{P}_{\Omega_k}(\boldsymbol{\Theta}^{t-1}))^{\top} \boldsymbol{v}(p) \tag{83}$$

    and transmit it to the central server.
7:     The central server forms the next iterate by $\boldsymbol{u}(p) = \sum_{k=1}^{K} \boldsymbol{u}_k(p)$.
8: **end for**
9: At the central server, compute the top left and right singular vector as $\boldsymbol{u}_{(1)}^t = \boldsymbol{u}(P)/\|\boldsymbol{u}(P)\|$ and $\boldsymbol{v}_{(1)}^t = \boldsymbol{v}(P)/\|\boldsymbol{v}(P)\|$.
10: **Return:** the top singular vectors $\boldsymbol{u}_{(1)}^t, \boldsymbol{v}_{(1)}^t$.

---

Line 4 and 5 in the above pseudo code implement the following power iterations:

$$\boldsymbol{v}(p) = \nabla_{\boldsymbol{\Theta}} \mathcal{L}(\boldsymbol{\alpha}^t, \boldsymbol{\Theta}^{t-1}) \boldsymbol{u}(p-1) = \sum_{k=1}^{K} \nabla_{\boldsymbol{\Theta}} \mathcal{L}_k(\boldsymbol{\alpha}^t, \mathcal{P}_{\Omega_k}(\boldsymbol{\Theta}^{t-1})) \boldsymbol{u}(p-1) \tag{84}$$

$$\boldsymbol{u}(p) = \nabla_{\boldsymbol{\Theta}} \mathcal{L}(\boldsymbol{\alpha}^t, \boldsymbol{\Theta}^{t-1})^{\top} \boldsymbol{v}(p) = \sum_{k=1}^{K} \nabla_{\boldsymbol{\Theta}} \mathcal{L}_k(\boldsymbol{\alpha}^t, \mathcal{P}_{\Omega_k}(\boldsymbol{\Theta}^{t-1}))^{\top} \boldsymbol{v}(p) , \tag{85}$$

where we have exploited the decomposable structure of the log-likelihood function in the distributed setting. Upon computing $\boldsymbol{u}_{(1)}^t, \boldsymbol{v}_{(1)}^t$, we can estimate the top singular value by $(\boldsymbol{v}_{(1)}^t)^{\top} \nabla_{\boldsymbol{\Theta}} \mathcal{L}(\boldsymbol{\alpha}^t, \boldsymbol{\Theta}^{t-1}) \boldsymbol{u}_{(1)}^t$ which can also be computed distributively using similar scheme as in the above. Consequently, the update direction $(\hat{\boldsymbol{\Theta}}^t, \hat{R}^t)$ can be computed at the central server using

$$(\hat{\boldsymbol{\Theta}}^t, \hat{R}^t) = \begin{cases} (\boldsymbol{0}, 0), & \text{if } \lambda_L \geq (\boldsymbol{v}_{(1)}^t)^{\top} \nabla_{\boldsymbol{\Theta}} \mathcal{L}(\boldsymbol{\alpha}^t, \boldsymbol{\Theta}^{t-1}) \boldsymbol{u}_{(1)}^t , \\ (-R_{\mathsf{UB}}^t \boldsymbol{u}_{(1)}^t (\boldsymbol{v}_{(1)}^t)^{\top}, R_{\mathsf{UB}}^t), & \text{if } \lambda_L < (\boldsymbol{v}_{(1)}^t)^{\top} \nabla_{\boldsymbol{\Theta}} \mathcal{L}(\boldsymbol{\alpha}^t, \boldsymbol{\Theta}^{t-1}) \boldsymbol{u}_{(1)}^t . \end{cases} \tag{86}$$

Lastly, to compute the step size $\beta_t$ required in line 6, an efficient way is to observe the following decomposition of the inner product:

$$\langle \boldsymbol{\Theta}^{t-1} - \hat{\boldsymbol{\Theta}}^t, \nabla_{\boldsymbol{\Theta}} \mathcal{L}(\boldsymbol{\alpha}^t, \boldsymbol{\Theta}^{t-1}) \rangle = \sum_{k=1}^{K} \langle \mathcal{P}_{\Omega_k}(\boldsymbol{\Theta}^{t-1} - \hat{\boldsymbol{\Theta}}^t), \nabla_{\boldsymbol{\Theta}} \mathcal{L}_k(\boldsymbol{\alpha}^t, \mathcal{P}_{\Omega_k}(\boldsymbol{\Theta}^{t-1})) \rangle . \qquad (87)$$

This implies that the inner product on the left hand side can be computed by aggregating the $K$ terms on the right hand side, where each of the $K$ terms can be computed at the $k$th worker once $\mathcal{P}_{\Omega_k}(\hat{\boldsymbol{\Theta}}^t)$ is available. As such, the central server also sends $\mathcal{P}_{\Omega_k}(\hat{\boldsymbol{\Theta}}^t)$ to the workers after (86). Consequently, the step size is given by:

$$\beta_t = \min \left\{ 1, \frac{(\hat{g}_{\boldsymbol{\Theta}}(\boldsymbol{\alpha}^t, \boldsymbol{\Theta}^{t-1}, R^{t-1}; R_{\mathsf{UB}}^t))_+}{\sigma_{\boldsymbol{\Theta}} \| \mathcal{P}_{\Omega}(\hat{\boldsymbol{\Theta}}^t - \boldsymbol{\Theta}^{t-1}) \|_F^2} \right\} , \qquad (88)$$

where

$$\hat{g}_{\boldsymbol{\Theta}}(\boldsymbol{\alpha}^t, \boldsymbol{\Theta}^{t-1}, R^{t-1}; R_{\mathsf{UB}}^t) := \langle \boldsymbol{\Theta}^{t-1} - \hat{\boldsymbol{\Theta}}^t, \nabla_{\boldsymbol{\Theta}} \mathcal{L}(\boldsymbol{\alpha}^t, \boldsymbol{\Theta}^{t-1}) \rangle + \lambda_L (R^{t-1} - \hat{R}^t) . \qquad (89)$$

Note that unlike the function $g_{\boldsymbol{\Theta}}(\cdot)$ defined in (49), the function $\hat{g}_{\boldsymbol{\Theta}}(\cdot)$ can be negative since the matrix $\hat{\boldsymbol{\Theta}}^t$ herein is computed from an inexact pair of top singular vectors.

Several remarks are in order. Throughout the optimization, the central server is unaware of the local gradient matrix *w.r.t.* $\boldsymbol{\Theta}$, instead only its corresponding matrix-vector products are transmitted from the workers to the server. In this way, the privacy-sensitive data from the workers will not be revealed to the server.

For any $\delta > 0$, it is well known that in high probability (with respect to the random initialization), the power method in Algorithm 2 converges [13] to an $\delta$-accurate top SVD solution in $P = \mathcal{O}(\log(1/\delta))$ steps[1]. Therefore, for the distributed MCGD method, the overall complexity required per iteration is $\mathcal{O}(|\Xi| + \max\{n, p\} \log(1/\delta))$ at the central server, and it is $\mathcal{O}(|\Omega_k| \max\{n, p\} \log(1/\delta))$ for the $k$th worker. The overall complexity is lower than a centralized implementation especially when $|\Omega_k| \ll |\Omega|$, e.g., when the number of workers increases.

$$\left| (\boldsymbol{u}_{(1)}^t)^\top \nabla_{\boldsymbol{\Theta}} \mathcal{L}(\boldsymbol{\alpha}^t, \boldsymbol{\Theta}^{t-1}) \boldsymbol{v}_{(1)}^t - \sigma_1(\nabla_{\boldsymbol{\Theta}} \mathcal{L}(\boldsymbol{\alpha}^t, \boldsymbol{\Theta}^{t-1})) \right| \leq \delta . \qquad (90)$$

In the complexity measure, we have hidden the dependency on the spectral gap $\Delta := \sigma_2(\nabla_{\boldsymbol{\Theta}} \mathcal{L}(\boldsymbol{\alpha}^t, \boldsymbol{\Theta}^{t-1})) / \sigma_1(\nabla_{\boldsymbol{\Theta}} \mathcal{L}(\boldsymbol{\alpha}^t, \boldsymbol{\Theta}^{t-1})) \leq 1$ in the big-O notation.