[Reviews · NeurIPS 2018]

Reviewer 1



The authors propose a new method to fit a low rank model to a data frame together with covariates or additive (eg row or column) effects. The authors show a guarantee on the results of their method, and give a simple and efficient algorithm for fitting the model. A few small remarks: * line 29: how do large vs small data frames differ wrt which methods are applicable? * terminology: you call this a low rank model, but instead consider a convex approach and fit a convex model, which need not satisfy any a priori rank bound. You should mention this earlier. * definition of X(k): how do you model column or row effects in this framework? As rank 1 matrices X(k)? If so, these seem very far from sparse, so that the ||X(k)||_1 term that enters the recovery bound is very large! * the connection with robust matrix completion is not clear, since generally both components are assumed to be unknown; whereas here it seems the X(k)s are given. * Theorem 1 is stated very cleanly and is easy to understand. It would be nice to have more interpretation of the bound, though, and suggestions for which parts are not tight. For example, the dependence on 1/pi seems like a gross overestimate; do you think this can be improved? * How big is ||X(k)||_1 usually? It seems like it would often be O(np), which would make your results more or less vacuous. * The setup for the numerical example with survey data should be clearer. What are X(k) here? How many are there? --- I have read the authors' rebuttal, and found it convincing. I have upgraded my score in light of (my new understanding of) the novelty of the theoretical guarantees and algorithm.

Reviewer 2



Summary ------- This paper introduces a new statistical model for matrices of heterogeneous data (data frames) based on the exponential family. The features of this model are: i) modeling additive effects in a sparse way, ii) modeling low-rank interactions. The parameters of this model are then estimated by maximizing the likelihood with sparse and low-rank regularizations. In addition, this work comes with statistical guarantees and optimization convergence guarantees of the proposed algorithm. Numerical experiments concludes the manuscript. Quality ------- This paper is mathematically rigorous and technically sound. Overall, this is a complete piece of work with theoretical guarantees (concerning the statistical aspect and the convergence of the proposed algorithm) and practical concerns (the estimation procedure is designed for large datasets). These two parts involve complex tools from statistics and optimization that seem completely understood by the authors. In the problem formulation (Section 2), the Gaussian case assumes known (observed) variance. Isn't that limited? Moreover, there are many hypotheses needed for the statistical guarantees. Is it common? Is it really necessary? The last section of the paper addresses two numerical experiments. The first one, on a synthetic dataset, honestly shows that the proposed method performs equally to a basic two-step procedure concerning estimating the low-rank parameter, but is beyond concerning the sparse parameter and the computational time (for large datasets). The second numerical experiment compares two versions of the proposed method (mixed effect versus Gaussian) about imputation error on a survey dataset. In both cases, I find the numerical evaluation quite basic and limited. What about more advanced methods? Previous works? As for me, this is a flaw of this paper (detailed below): the link with the related work is unclear. Clarity ------- This submission is a readable paper about a subject I am not familiar with. I find it very clear and polished (the only typos I saw is "match" L. 230 and \lambda_1 L. 157). The authors made a noticeable effort to explain the implications of hypotheses (Section 2). But what about their concrete meaning? For instance, how restrictive is the last part of H1? What about positive definiteness of G in H3? Which distributions satisfy H4? More importantly, I do not understand the aim of H5 since, by the model, Y_ij are assumed distributed according to an exponential family. Moreover, Theorem 1 is not very clear to me: does it prove consistency? When \pi goes to 1, one expects the estimators to be consistent, but it is not clear with the proposed bounds. Some references are missing L. 162 and 213. Originality ----------- This work is a novel and smart combination of well-known techniques, concerning both statistics and optimization. However, it is not clear how it differs from previous studies. In particular, a section concerning related work is missing (the optimization part is briefly linked to prior work P. 6 but this effort is missing for the statistical part). I agree that the page limitation makes the thing difficult for the authors, yet the link to prior work is needed to assess the paper. For instance, this submission seems quite supported by [23], but how it is related? Concretely, what is new in the statistical model (3)? This remark also holds for the (statistics and optimization) proofs: are they minor modifications of existing content or original work? In addition, this lack also holds for the numerical section, where the proposed approach is only compared to either a basic procedure or itself. Significance ------------ As for me, the work described in this submission is a significant contribution to the field of data frame analysis, both by its statistical and optimization aspects. It deserves publication as soon as the link to the prior work is made clearer and the numerical evaluation is strengthened by competitive advanced methods. Rebuttal ----------- I have read the rebuttal and I thank the authors for their clear response. The authors agreed to add a "Related work" section, clarifying the link to prior work and originality of the proposed study. They also answered convincingly the reviewers' questions. Consequently, I have upgraded the overall score.

Reviewer 3



Summary ------- The paper introduces a low rank + sparse additive effect model to jointly estimate main effects and interactions. Statistical guarantees are derived and an optimization algorithm combining Frank-Wolfe and coordinate descent is proposed. The paper seems technically sound, well written and overall easy to follow. Regarding statistical guarantees, the authors mention [27, 4, 5, 13] (noiseless setting) and [17] (additive noise setting) as prior analyses but it's not clear what's the difference with the proposed analysis. I would explicitly explain the difference right from line 85. Unfortunately, the numerical experiments were quite limited. The first experiment (Table 1) is essentially a sanity check on data generated from the model. The second experiment is interesting but only compares two variants of the proposed approach and there is no comparison with other baselines. Overall, this is a fine paper but the experiments are too lacking to make it a clear accept. Detailed comments ----------------- * Line 56: often referred to as Frobenius inner product * Line 81: Could you briefly explain in what way LORIS generalizes [4], so that the reader doesn't need to read [4] to figure it out. * I believe the work below could be used to solve a constrained version of (9) Frank-Wolfe Splitting via Augmented Lagrangian Method Gauthier Gidel, Fabian Pedregosa and Simon Lacoste-Julien. Proceedings of the 21st International Conference on Artificial Intelligence and Statistics (AISTATS), 2018. * Figure 1: the labels are too small * Line 294: what do you mean by mixed data model here? could you be more explicit, perhaps give the exact model specification? * I feel the paper doesn't discuss related work sufficiently. The discussion in the introduction is quite limited. The authors should spend more time explaining what's missing from the existing works and why it's important. The authors could create a related work section after the introduction, move the solver comparison ("Comparison to Prior Work") there and add more discussion on existing low-rank + sparse models and their statistical guarantees.